# REL-RAG: RELATION-AWARE RETRIEVAL-AUGMENTED GENERATION FOR GENERALIZABLE KNOWLEDGE GRAPH QUESTION ANSWERING

## ABSTRACT

Large Language Models (LLMs) augmented with knowledge graphs (KGs) have been widely studied for knowledge graph question answering (KGQA). Graph-based retrievers exhibit strong empirical performance, but their generalization ability remains limited. In this work, we show that applying a *line graph transformation* to the KG provably enhances the generalizability of GNN-based retrievers. By elevating relations to first-class objects, line graphs encode relation transitions explicitly, and the resulting inductive bias aligns naturally with relational reasoning in KGs. This alignment makes multi-hop reasoning substantially easier to learn and improves generalizability across different types of distribution shifts. Building upon this representation, we propose `REL-RAG`, a framework that emphasizes relational reasoning for graph retrievers and is equipped with two complementary training objectives for flexible integration with LLMs. Path-based learning achieves higher precision with fewer tokens, making it especially suitable for smaller LLMs with limited context capacity. Triple-based learning encourages richer evidence diversity, which stronger LLMs can exploit more effectively with larger token budgets. Empirically, `REL-RAG` establishes new state-of-the-art results on KGQA benchmarks, surpassing prior graph retrievers by up to $18.10\%$ with Llama3.1-8B and $10.63\%$ with GPT-4o.

## 1 INTRODUCTION

Large Language Models (LLMs) (Brown et al., 2020; Achiam et al., 2023; Touvron et al., 2023) have demonstrated remarkable capabilities in complex reasoning tasks across various domains (Wu et al., 2024; Fan et al., 2024; Manning et al., 2024), marking a significant step toward bridging the gap between human cognition and artificial general intelligence (AGI) (Huang & Chang, 2023; Wei et al., 2022; Yao et al., 2024; Bubeck et al., 2023). However, the reliability of LLMs remains a pressing concern due to outdated knowledge (Kasai et al., 2023) and hallucination (Ji et al., 2023; Huang et al., 2023). These issues severely undermine their trustworthiness in knowledge-intensive applications.

To mitigate these deficiencies, Retrieval-Augmented Generation (RAG) (Gao et al., 2024b; Lewis et al., 2020) has been introduced to ground LLMs with external knowledge. While effective, most existing RAG pipelines rely on unstructured text corpora, which are often noisy, redundant, and semantically diffuse (Shuster et al., 2021; Gao et al., 2024b). In contrast, Knowledge Graph (KG) (Hogan et al., 2021) organizes information as structured triples $(h, r, t)$, providing a compact and semantically rich representation of real-world facts (Chein & Mugnier, 2008; Robinson et al., 2015). As a result, incorporating KGs into RAG frameworks (i.e., KG-based RAG) has emerged as a vibrant and evolving area for achieving faithful and interpretable reasoning.

Building upon the KG-based RAG frameworks, recent studies have proposed methods that combine KGs with LLMs for Knowledge Graph Question Answering (KGQA) (Wang et al., 2023; Dehghan et al., 2024; Mavromatis & Karypis, 2022; Luo et al., 2024; Mavromatis & Karypis, 2024; Chen et al., 2024; Li et al., 2024). A prevalent approach among these methods is the *retrieve-then-reasoning* paradigm, where a retriever first extracts relevant knowledge from the KG, and subsequently an LLM-based reasoner generates answers based on the retrieved information. The retriever can be roughly categorized into LM-based retriever and graph-based retriever. Notably, recent studies

have demonstrated that graph neural network (GNN (Kipf & Welling, 2016; Hamilton et al., 2017; Velickovic et al., 2017; Xu et al., 2019))-based graph retrievers can achieve superior performance in KGQA tasks, even without fine-tuning the LLM reasoner (Mavromatis & Karypis, 2022; Li et al., 2024). Despite these successes, recent studies also reveal generalization limitations of graph retrievers: the cross-dataset performance drops substantially relative to in-distribution evaluation, indicating limited robustness to distribution shift (Li et al., 2024). This raises the research question:

*How to enhance the generalizability of graph retrievers for KGQA tasks?*

In this work, we identify **relational mixing** as a key reason for the limited generalization of graph retrievers: entity nodes aggregate messages from all incident relations, making it challenging to learn ground-truth multi-hop reasoning patterns. To address this, we propose applying *line graph transformation* to the KG, which provably improves the generalization ability of graph retrievers without altering their architectures. The key insight is that **reasoning over relation compositions becomes fundamentally easier when relations are elevated to first-class objects** (i.e., relations are elevated from implicit edge attributes to explicit nodes that can be directly operated upon during message passing), as the structural bias induced by line graphs aligns naturally with relational reasoning in KGs, thereby facilitating multi-hop reasoning and strengthening generalization across datasets.

To illustrate, consider a simple case where a 2-hop relation path $r_1 \rightarrow r_2$ provides the correct evidence for answering a question $q$. Learning the representation $h^*_{r_1 r_2}$ for this path is difficult under standard message-passing on the raw KG: the entity node aggregates information from all incident relations, entangling relational semantics and making $h^*_{r_1 r_2}$ hard to capture. In contrast, in the line graph, the composition $(r_1, r_2)$ corresponds to two adjacent triple-nodes, where the relation transition is explicitly encoded in the topology. This introduces a structural bias that makes $h^*_{r_1 r_2}$ substantially easier to learn (see Sec. 3 and Figure 1 for more details). More formally, we prove that line-graph models admit tighter generalization bounds than their entity-graph counterparts across multiple types of distribution shift.

While the line-graph transformation addresses the structural limitations, we still need to address how to make the RAG pipeline operate under different LLM capacities and token budgets. To this end, we propose REL-RAG, a framework that emphasizes relational reasoning for KGQA, and equips graph retrievers with two complementary training regimes to flexibly adapt to different reasoning capacities of downstream LLMs: **(1)** Path-based learning, which achieves higher precision with fewer tokens, making it suitable for smaller LLM reasoners with limited reasoning ability. **(2)** Triple-based learning, which emphasizes evidence diversity, better suited for stronger LLM reasoners capable of processing larger token budgets.

Our contributions are summarized as follows:

- We formally demonstrate that applying a line graph transformation can provably enhance the generalizability of GNN-based retrievers under various types of distribution shift.

- We propose REL-RAG, a framework that emphasizes relational reasoning for graph retrievers, offering flexible integration with LLMs through two complementary training regimes for generalizable RAG: path-based learning for token-efficient reasoning with smaller LLMs, and triple-based learning for evidence-diverse reasoning with stronger LLMs.

- REL-RAG establishes new state-of-the-art results on KGQA benchmarks, surpassing prior graph retrievers by up to $18.10\%$ with Llama3.1-8B and $10.63\%$ with GPT-4o.

## 2 PRELIMINARY

**triple $\tau$.** A triple represents a factual statement: $\tau = \langle e, r, e' \rangle$, where $e, e' \in \mathcal{E}$ denote the subject and object entities, respectively, and $r \in \mathcal{R}$ represents the relation linking these entities.

**Reasoning Path $p$.** A reasoning path $p := e_0 \xrightarrow{r_1} e_1 \xrightarrow{r_2} \cdots \xrightarrow{r_k} e_k$ connects a source entity to a target entity through one or more intermediate entities. Moreover, we denote $z_p := \{r_1, r_2, \cdots r_k\}$ as the relation path of $p$.

Figure 1: Illustration of a raw knowledge graph and its line-graph transformation. In the raw graph, learning the ground-truth representation of a 2-hop path is challenging due to relation mixing. In contrast, the representation of a path to be easily learned under the MPNN framework by gating the message flow along the corresponding edges, with line graph transformation.

**Problem setup.** Given a natural language question $q$ and a knowledge graph $\mathcal{G}$, our goal in this study is to learn a function $f_\theta$ that takes as inputs the question entity $e_q$, and a subgraph $\mathcal{G}_q \subset \mathcal{G}$, to infer an answer entity $e_a \in \mathcal{G}_q$. Following previous practice, we assume that $e_q$ are correctly identified and linked in $\mathcal{G}_q$.

Next, following (Gu et al., 2021), we formally define three types of generalization challenge in KGQA, followed by the definition of *directed line graph*.

**Definition 1** (In-Distribution Generalization). Given training and test distributions $\mathcal{D}_{\text{train}} = \mathcal{D}_{\text{test}}$ over question-answer pairs, a retriever $f_\theta$ exhibits In-Distribution (ID) generalization when it maintains performance on test samples where all entities $\mathcal{E}_{\text{test}} \subseteq \mathcal{E}_{\text{train}}$, relations $\mathcal{R}_{\text{test}} \subseteq \mathcal{R}_{\text{train}}$, and relation paths $\mathcal{Z}_{\text{test}} \subseteq \mathcal{Z}_{\text{train}}$ have appeared during training.

**Definition 2** (Compositional Generalization). A retriever demonstrates compositional generalization when it correctly processes reasoning paths containing novel relation compositions. Formally, for $z_p = \{r_1, \ldots, r_k\}$ where each $r_i \in \mathcal{R}_{\text{train}}$, but there exists a subsequence $(r_i, \ldots, r_j)$ such that this specific composition was not observed during training, i.e., $(r_i, \ldots, r_j) \notin \mathcal{Z}_{\text{train}}$.

**Definition 3** (Out-of-Distribution Generalization). Out-of-Distribution (OOD) generalization occurs when the test distribution contains novel elements: $\mathcal{E}_{\text{test}} \nsubseteq \mathcal{E}_{\text{train}}$ or $\mathcal{R}_{\text{test}} \nsubseteq \mathcal{R}_{\text{train}}$. The retriever must leverage semantic similarities between seen and unseen elements to maintain performance despite encountering entities or relations absent from training.

**Definition 4.** (*Directed Line Graph*) Given a directed graph $\mathcal{G} = (\mathcal{V}, \mathcal{E})$, where each edge $e = (u, v) \in \mathcal{E}$ has a direction from $u$ to $v$, the *directed line graph* $l(\mathcal{G})$ is a graph where:

- Each node in $l(\mathcal{G})$ corresponds to a directed edge in $\mathcal{G}$.

- There is a directed edge from node $e_1 = (u, v)$ to node $e_2 = (v, w)$ in $l(\mathcal{G})$ if and only if the target of $e_1$ matches the source of $e_2$ (i.e., $v$ is shared and direction is preserved).

With the line graph transformation, each triple $(h, r, t)$ is reified as a node, elevating relations to first-class objects. Next, we show that this structural bias explicitly encodes relation transitions and thereby facilitates the generalization of graph retrievers.

## 3 WHY LINE GRAPH ENHANCES GENERALIZABILITY OF GRAPH RETRIEVERS?

Prior work highlights the importance of relation paths for generalization in knowledge graphs (Sun et al., 2024a; Luo et al., 2024; Galkin et al., 2023; Geng et al., 2023; Lee et al., 2023; Gao et al., 2023; Zhou et al., 2023). Yet, on the entity graph (raw graph) each node aggregates from all incident relations, entangling relational semantics and obscuring multi-hop patterns. Next, we present a simple case study by contrasting a generic two-layer Message Passing Neural Network (MPNN) on the raw KG with an MPNN on its line graph counterpart.

**MPNN on raw KGs.** Let $h_v^{(l)} \in \mathbb{R}^d$ be the representation of entity $v$ at layer $l$, and let $r_{uv}$ be the relation on edge $(u, v)$. A general MPNN layer is

$$m_{v \leftarrow u}^{(l)} = \psi^{(l)}\big(h_v^{(l)}, h_u^{(l)}, r_{uv}\big), \qquad h_v^{(l+1)} = \phi^{(l)}\Big(h_v^{(l)}, \square_{u \in \mathcal{N}(v)} m_{v \leftarrow u}^{(l)}\Big), \tag{1}$$

where $\square$ is any permutation-invariant aggregator, $\psi^{(l)}$ is the layer-$l$ message function, and $\phi^{(l)}$ is the layer-$l$ update function that combines $h_t^{(l)}$ with the aggregated messages to yield $h_t^{(l+1)}$.

For a 2-hop pattern $u \xrightarrow{r_1} v \xrightarrow{r_2} w$, the representation at $w$ after two layers unfolds as

$$h_w^{(2)} = \phi^{(1)}\Big(h_w^{(1)}, \square_{v \in \mathcal{N}(w)} \psi^{(1)}\big(h_w^{(1)}, \underbrace{\phi^{(0)}(h_v^{(0)}, \square_{u \in \mathcal{N}(v)} \psi^{(0)}(h_v^{(0)}, h_u^{(0)}, r_{uv}))}_{h_v^{(1)}}, r_{vw}\big)\Big). \tag{2}$$

Thus it is easy to see that isolating the target representation $h_{r_1 r_2}^*$ requires demixing many coupled terms, which is hard and non-trivial.

**MPNN on line graph.** Form the directed line graph whose nodes are triples $t = (e, r, e')$; there is an edge $t \to t'$ iff the tail of $t$ equals the head of $t'$ and direction is preserved. Let $z_t^{(l)}$ be the triple-node representation. A general MPNN layer on the line graph is

$$m_{t \leftarrow t'}^{(l)} = \psi^{(l)}\big(z_t^{(l)}, z_{t'}^{(l)}\big), \qquad z_t^{(l+1)} = \phi^{(l)}\Big(z_t^{(l)}, \square_{t' \in \mathcal{N}(t)} m_{t \leftarrow t'}^{(l)}\Big). \tag{3}$$

To learn the 2-hop relation representation $h_{r_1 r_2}^*$, it suffices to gate messages by the relation pair:

$$m_{t \leftarrow t'}^{(l)} = \psi^{(l)}\big(z_t^{(l)}, z_{t'}^{(l)}\big) \cdot \mathbf{1}\big[r(t) = r_1\big] \cdot \mathbf{1}\big[r(t') = r_2\big], \tag{4}$$

so the solution is that only valid $(r_1 \to r_2)$ transitions contribute and all other neighbor messages are zero. Consequently, the target representation is obtained by a single, pair-specific message flow:

$$z_t^{(1)} = \phi^{(0)}\Big(z_t^{(0)}, \square_{t' \in \mathcal{N}(t):\, r(t) = r_1,\, r(t') = r_2} \psi^{(0)}\big(z_t^{(0)}, z_{t'}^{(0)}\big)\Big), \tag{5}$$

making $h_{r_1 r_2}^*$ easy to spot and learn without demixing. These observations explain why the line-graph transformation induces a beneficial structural bias for multi-hop reasoning. Figure 1 provides an intuitive illustration of how the line-graph transformation disentangles relation transitions for multi-hop reasoning. As shown, the line graph requires only truncating a single erroneous message flow (marked by scissors) to recover the ground-truth 2-hop reasoning path, whereas the relation mixing in the entity graph significantly complicates optimization.

The following theorems formally establish that the line-graph transformation improves generalization under various distribution shifts. All the proofs can be found in Appendix A.

**Proposition 1** (Bijective Mapping of Directed Paths). *Let* $P = \Big(e_0 \xrightarrow{r_1} e_1 \xrightarrow{r_2} \cdots \xrightarrow{r_k} e_k\Big)$, *where* $k \geq 1$, *be a directed path of length* $k$ *in* $\mathcal{G}$, *and each step is the directed edge (triple)* $t_i = \langle e_{i-1}, r_i, e_i \rangle \in \mathcal{E}$ *for* $i = 1, \ldots, k$. *Define the mapping* $\Phi(P) = (t_1 \to t_2 \to \cdots \to t_k)$, *where each* $t_i$ *is regarded as a node in the directed line graph* $l(\mathcal{G})$. *Then* $\Phi$ *defines a bijection between the set of directed paths of length* $k$ *in* $\mathcal{G}$ *and the set of directed paths of length* $k - 1$ *in* $l(\mathcal{G})$.

This bijective mapping ensures that no path information is lost during transformation, providing a solid foundation for analyzing the generalization properties of models operating on line graphs. We now present theoretical guarantees showing that line graph representations lead to improved generalization bounds across all three generalization scenarios.

**Theorem 3.1** (ID Generalization Bound). *Let* $d$ *denote the embedding dimension and* $m$ *the number of i.i.d. training samples. The Rademacher complexity of models operating on the entity graph* $\mathcal{G}$ *and its line graph* $\mathcal{G}' = l(\mathcal{G})$ *satisfy:*

$$\Re_m(\mathcal{H}_{\mathcal{G}}) = O\left(\frac{\sqrt{R}d}{\sqrt{m}}\right), \qquad \Re_m(\mathcal{H}_{\mathcal{G}'}) = O\left(\frac{d}{\sqrt{m}}\right). \tag{6}$$

*Then, with probability at least $1 - \delta$ over an i.i.d. sample of size $m$, the following generalization bounds hold:*

$$\mathcal{L}(h_{\mathcal{G}}) \leq \hat{\mathcal{L}}_m(h_{\mathcal{G}}) + O\left(\frac{\sqrt{R}d}{\sqrt{m}}\right) + \sqrt{\frac{\log(1/\delta)}{2m}},$$

$$\mathcal{L}(h_{\mathcal{G}'}) \leq \hat{\mathcal{L}}_m(h) + O\left(\frac{d}{\sqrt{m}}\right) + \sqrt{\frac{\log(1/\delta)}{2m}}.$$

Theorem 3.1 implies that the retrieval model operating on line graph $\mathcal{G}'$ reduces the estimation term by a factor of $\sqrt{R}$ relative to the entity graph, hence the sample complexity required to achieve a given excess risk reduces by a factor of $\sqrt{R}$ on $\mathcal{G}'$, indicating improved ID generalization. Next, we provide the theoretical results on compositional shift and OOD shift.

**Theorem 3.2** (Compositional Shift Bound). *Let $h_{\mathcal{G}'}$ be the predictor produced by a line-graph GCN and consider an* unseen *ordered pair of relations $(r_1, r_2)$. Let $\mathcal{D}_S$ and $\mathcal{D}_T$ denote the training and test distributions, respectively, and let $m$ be the number of i.i.d. training samples. Then, with probability at least $1 - \delta$ over the draw of the training set, the PAC-Bayesian generalization bound specializes to*

$$\mathcal{L}_{\mathcal{D}_T}(h_{\mathcal{G}'}) \leq \underbrace{\mathcal{L}_{\mathcal{D}_S}(h_{\mathcal{G}'}) + \sqrt{\frac{KL(Q\|P) + \log(2\sqrt{m}/\delta)}{2m}}}_{\text{standard PAC-Bayes term}} + \epsilon_{shift}^{l(\mathcal{G})}, \tag{7}$$

*where the distribution-shift terms for the line-graph and entity-graph models satisfy*

$$\epsilon_{shift}^{l(\mathcal{G})} = O\left(\sqrt{\frac{d}{\min\{N(r_1),\ N(r_2)\}}}\right), \qquad \epsilon_{shift}^{\mathcal{G}} = O\left(\sqrt{\frac{R\,d}{\min\{N(r_1),\ N(r_2)\}}}\right). \tag{8}$$

*Here, $N(r)$ is the number of training instances involving relation $r$, $d$ is the embedding dimension, and $R$ is the number of relations.*

The line graph removes the $\sqrt{R}$ penalty as in the ID cases. However, Theorem 3.2 involves a data-dependent term: the shift term $\epsilon_{shift}$ scales as $1/\sqrt{\min N(r_1), N(r_2)}$, so rare relations constrain robustness even under the line graph's favorable structural bias, which poses a more challenging scenario than in-distribution learning.

**Theorem 3.3** (OOD Generalization Bound). *Let $r_{new} \notin \mathcal{R}_{train}$ be an unseen relation and let $\Delta = \min_{r \in \mathcal{R}_{train}} d(r_{new}, r)$ with $r_{sim} = \arg\min_{r \in \mathcal{R}_{train}} d(r_{new}, r)$. Under the same confidence event as in Theorem 3.2, the line-graph predictor $h_{\mathcal{G}'}$ and entity-graph predictor $h_{\mathcal{G}}$ obeys the bound equation 7 with*

$$\epsilon_{shift}^{l(\mathcal{G})} = O\left(L\,\Delta + \sqrt{\frac{d}{N(r_{sim})}}\right), \qquad \epsilon_{shift}^{\mathcal{G}} = O\left(L\,\Delta + \sqrt{\frac{R\,d}{N(r_{sim})}}\right). \tag{9}$$

*Here, $L$ is the Lipschitz constant of the respective predictor with respect to the relation representation metric $d(\cdot, \cdot)$, $N(r_{sim})$ is the number of training examples containing the most semantically similar seen relation $r_{sim}$, $d$ is the embedding dimension, and $R$ is the number of relations.*

Theorem 3.3 follows the same proof process as Theorem 3.2, decomposing the PAC-Bayesian in-distribution term plus an explicit shift term $\epsilon_{\text{shift}}$. Beyond the data coverage $N(r_{\text{sim}})$, OOD shift is further limited by the semantic proximity $\Delta$ to the nearest seen relation, making OOD generalization more challenging than the ID and compositional cases. In all cases, operating on the line graph replaces the entity-graph factor $\sqrt{R\,d}$ by $\sqrt{d}$, yielding a $\sqrt{R}$ reduction in the shift term and thus tighter guarantees. In summary, this section has established the theoretical foundation for why line graph representations enhance the generalizability of graph retrievers in KBQA tasks. This structural advantage provides a principled basis for building more robust retrieval systems. Building on this line graph representation, we next present two distinct learning regimes that leverage these benefits while adapting to practical constraints of different LLM reasoners and token limits.

# 4 TWO FLEXIBLE LEARNING REGIMES

In this section, we investigate two learning regimes within `REL-RAG`, which is built upon the line-graph representation. These objectives emphasize different trade-offs between precision and diversity, enabling `REL-RAG` to flexibly adapt to different classes of LLM reasoners.

## 4.1 PATH-BASED LEARNING

Beyond the findings and contributions presented throughout this work, the path-based learning regime represents our main technical innovation in `REL-RAG`. Although prior work has adopted path-based paradigms, these approaches typically use reasoning paths in a training-free manner (Sun et al., 2024a; Xu et al., 2024b;a; Chen et al., 2024; Li et al., 2025; Liang & Gu, 2025) or only incorporate paths during inference (Mavromatis & Karypis, 2024; Li et al., 2024). In contrast, we propose a path-based learning method for neural retrievers, which proves effective when paired with small LLMs.

The proposed path-based learning models reasoning as a sequential prediction process over the line graph $\mathcal{G}'_q$, where each node $v_i$ represents a triple $(e_i, r_i, e'_i)$. Given a question $q$ and a reasoning path $(v_{q(0)}, v_{q(1)}, \ldots, v_{q(K-1)})$ that connects the question triple $v_{q(0)}$ to the answer triple $v_{q(K-1)}$, the objective is:

$$\max_\theta \ \mathbb{P}_\theta \left( v_{q(i)} \,\middle|\, v_{q(0)}, \ldots, v_{q(i-1)}, q, \mathcal{G}'_q \right), \quad i \in [1, K], \tag{10}$$

where $\theta$ are the parameters of the graph retriever.

**Training.** The node representation for each $v_i$ in $\mathcal{G}'_q$ is obtained through: $\mathbf{z}_i = f_\theta(v_i; \mathcal{G}'_q)$, where $f_\theta(\cdot)$ consists of two 2-layer MPNN models, one operating on $\mathcal{G}'_q$ and the other on its edge-reversed counterpart, followed by summation to obtain the final representation of $\mathbf{z}_i$. The path selection at each step $i$ is optimized via:

$$\mathcal{L}_{\text{path}} = \mathbb{E}_\mathcal{D} \left[ - \log \frac{\exp(\langle \mathbf{z}_q, \mathbf{z}_{q(i)} \rangle)}{\sum_{j \in \mathcal{N}(q(i-1))} \exp(\langle \mathbf{z}_q, \mathbf{z}_j \rangle)} \right], \quad i > 0, \tag{11}$$

where $\mathbf{z}_q$ is the question representation and $\mathcal{N}(q(i-1))$ denotes the neighbor set of the previously selected node $v_{q(i-1)}$ in $\mathcal{G}'_q$, augmented with a special stop node to indicate when sampling should terminate. The stop node representation is computed as $\mathbf{z}_{\text{stop}} = \sum_{k=0}^{i-1} \mathbf{z}_{q(k)} + \mathbf{z}_q$. When multiple valid paths exist, we randomly select one to optimize Eqn. 11.

**Selecting the initial triple.** At step $i = 0$, the model must select the question triple $v_{q(0)}$ from candidates involving the question entity $e_q$:

$$\mathcal{V}_{\text{cand}} := \{ v_i \mid v_i = (e_i, r_i, e'_i), \ e_i = e_q \}. \tag{12}$$

One natural choice would be to treat only the ground-truth $v_{q(0)}$ as positive and all other candidates as negative. However, we find this approach hurts performance because certain nodes in $\mathcal{V}_{\text{cand}}$ share semantic similarities with $v_{q(0)}$, such as having similar relations or targeting related entities. Treating these semantically related triples as negative samples introduces noise during training and impairs the model's ability to learn meaningful representations. To address this issue, we propose positive sample augmentation. Specifically, an auxiliary LLM (GPT-4o-mini in our experiments) analyzes the question $q$ and selects a set of probable relations $\mathcal{R}_*$ that are semantically relevant to the query. The augmented positive set is then defined as:

$$\mathcal{V}_{\text{pos}} := \{ v_{q(0)} \} \cup \{ v_i \mid v_i = (e_i, r_i, e'_i), e_i = e_q, r_i \in \mathcal{R}_* \}. \tag{13}$$

The initial selection is optimized via negative sampling:

$$\mathcal{L}_q = - \sum_{v^+ \in \mathcal{V}_{\text{pos}}} \log \sigma(\langle \mathbf{z}_q, \mathbf{z}_{v^+} \rangle) + \sum_{v^- \in \mathcal{V}_{\text{neg}}} \log \sigma(-\langle \mathbf{z}_q, \mathbf{z}_{v^-} \rangle), \tag{14}$$

where $\mathcal{V}_{\text{neg}} = \mathcal{V}_{\text{cand}} \setminus \mathcal{V}_{\text{pos}}$ and $\sigma(\cdot)$ is the sigmoid function.

## 4.2 TRIPLE-BASED LEARNING

Triple-based learning relaxes the sequential constraint of path-based learning and instead optimizes over sets of triples independently. Similar to path-based learning, we employ negative sampling where the positive set $\mathcal{V}'_{\text{pos}}$ contains all triples appearing in the annotated reasoning paths, while the negative set $\mathcal{V}'_{\text{neg}}$ comprises the remaining nodes in $\mathcal{G}'_q$.

The triple-based learning objective is formulated as:

$$\mathcal{L}_{\text{triple}} = \mathbb{E}_{q \sim \mathcal{D}} \left[ - \sum_{v^+ \in \mathcal{V}'_{\text{pos}}} \log \sigma(\langle \mathbf{z}_q, \mathbf{z}_{v^+} \rangle) + \sum_{v^- \in \mathcal{V}'_{\text{neg}}} \log \sigma(-\langle \mathbf{z}_q, \mathbf{z}_{v^-} \rangle) \right], \quad (15)$$

where $\mathbf{z}_q$ and $\mathbf{z}_v$ represent the question and triple embeddings respectively, obtained through the same GNN architecture as in path-based learning.

## 4.3 EMPIRICAL OBSERVATIONS

Our experiments reveal distinct advantages for each learning regime. Path-based learning yields more accurate retrieved evidence with correct paths occurring at higher probabilities, requiring fewer triples for effective reasoning. This approach is particularly suitable for medium-scale LLMs (e.g., 7B parameters), which often struggle to identify relevant facts within large, noisy contexts. The sequential nature of path-based retrieval naturally reduces distractors and improves interpretability.

Conversely, triple-based learning excels when paired with high-end LLMs. These models can effectively process larger contexts and benefit from the broader coverage of candidate facts. The increased diversity compensates for the additional noise, as stronger models demonstrate superior capability in extracting relevant evidence from complex contexts. This flexibility allows `REL-RAG` to adapt its retrieval strategy based on the downstream LLM's capacity, optimizing the precision-diversity trade-off accordingly.

## 5 EXPERIMENTS

In this section, we first detail the experimental setup, we then evaluate in-distribution performance, followed by cross-dataset results that probe compositional and OOD shifts. Next, we present a case study of `REL-RAG` on the line-graph and raw graph representations to illustrate the retrieved evidence. Finally, we report an ablation study on retriever architectures.

Additionally, we provide efficiency analysis, more ablation studies, and examples of retrieved reasoning paths and triples in Appendix F.

### 5.1 EXPERIMENT SETUP

**Datasets.** We conduct experiments on three widely used and challenging benchmarks for KGQA. (1) **WebQSP** (Yih et al., 2016) and (2) **CWQ** (Talmor & Berant, 2018), both constructed to evaluate multi-hop reasoning capabilities, with questions requiring up to four hops of inference over a large-scale knowledge graph. The underlying knowledge base for both datasets is Freebase (Bollacker et al., 2008). (3) **GrailQA** (Gu et al., 2021) covers domains of questions that differ from those in WebQSP and

Table 1: Test performance on WebQSP and CWQ. Best results are **bold**; second-best are underlined. *Ours(P)* denotes path-based training, *Ours(T)* denotes triple-based training.

| | Method | WebQSP | | CWQ | |
|---|---|---|---|---|---|
| | | Macro-F1 | Hit | Macro-F1 | Hit |
| LLM | Qwen-7B (Yang et al., 2024) | 35.5 | 50.8 | 21.6 | 25.3 |
| | Llama3.1-8B (Meta, 2024) | 34.8 | 55.5 | 22.4 | 28.1 |
| | GPT-4o-mini (OpenAI, 2024) | 40.5 | 63.8 | 40.5 | 63.8 |
| | ChatGPT (OpenAI, 2022) | 43.5 | 59.3 | 30.2 | 34.7 |
| | ChatGPT+CoT (Wei et al., 2022) | 38.5 | 73.5 | 31.0 | 47.5 |
| KG+LLM | UniKGQA (Jiang et al., 2022) | 72.2 | – | 49.0 | – |
| | KD-CoT (Wang et al., 2023) | 52.5 | 68.6 | – | 55.7 |
| | ToG (GPT-4) (Sun et al., 2024a) | – | 82.6 | – | 67.6 |
| | StructGPT (Jiang et al., 2023) | – | 74.6 | – | – |
| | Retrieve-Rewrite-Answer (Wu et al., 2023) | – | 79.3 | – | – |
| | G-Retriever (He et al., 2024) | 53.4 | 73.4 | – | – |
| | RoG (Luo et al., 2024) | 70.2 | 86.6 | 54.6 | 61.9 |
| | EtD (Liu et al., 2024) | – | 82.5 | – | 62.0 |
| | GNN-RAG (Mavromatis & Karypis, 2024) | 71.3 | 85.7 | 59.4 | 66.8 |
| | SubgraphRAG + Llama3.1-8B (Li et al., 2024) | 70.5 | 86.6 | 47.2 | 56.9 |
| | SubgraphRAG + GPT-4o-mini (Li et al., 2024) | 77.4 | 90.1 | 54.1 | 62.0 |
| | SubgraphRAG + GPT-4o (Li et al., 2024) | 76.4 | 89.8 | 59.1 | 66.6 |
| | SubgraphRAG + GPT-4o-mini (500) (Li et al., 2024) | 77.6 | 91.2 | 55.4 | 64.9 |
| 50 Triples | Ours(P) + Llama3.1-8B | 77.3 | 88.8 | 56.8 | 67.2 |
| | Ours(T) + Llama3.1-8B | 72.6 | 90.2 | 53.6 | 65.6 |
| | Ours(P) + GPT-4o-mini | 80.4 | 92.5 | 58.1 | 69.3 |
| | Ours(T) + GPT-4o-mini | 78.7 | 92.5 | 58.6 | 68.3 |
| | Ours(P) + GPT-4o | **80.7** | 93.3 | 58.8 | 69.0 |
| | Ours(T) + GPT-4o | 79.6 | 93.1 | 58.2 | 67.9 |
| 500 Triples | Ours(P) + Llama3.1-8B | 74.9 | 92.1 | 52.9 | 67.2 |
| | Ours(T) + Llama3.1-8B | 73.6 | 91.2 | 55.1 | 66.1 |
| | Ours(P) + GPT-4o-mini | 80.4 | 93.3 | 56.1 | 67.6 |
| | Ours(T) + GPT-4o-mini | 79.9 | **94.0** | 61.3 | 71.6 |
| | Ours(P) + GPT-4o | 79.9 | 92.8 | 56.8 | 68.1 |
| | Ours(T) + GPT-4o | 80.2 | 93.6 | **61.5** | **71.8** |

Table 2: Cross-dataset generalization with training on WebQSP and evaluation on WebQSP/CWQ/-GrailQA. Best results are **bold**; second-best are underlined. Both methods use triple-based learning with 500 triples. The LLM reasoner is GPT-4o-mini.

| Method | WebQSP | | CWQ | | GrailQA | |
|---|---|---|---|---|---|---|
| | Macro-F1 | Hit | Macro-F1 | Hit | Macro-F1 | Hit |
| Raw graph | 74.32 | 91.08 | 51.78 | 62.84 | 32.99 | 49.67 |
| Line graph | **78.67** | **92.49** | **57.00** | **67.52** | **34.68** | **52.42** |
| *Relative gain vs. Raw* | +5.9% | +1.6% | +10.1% | +7.5% | +5.12% | +5.54% |

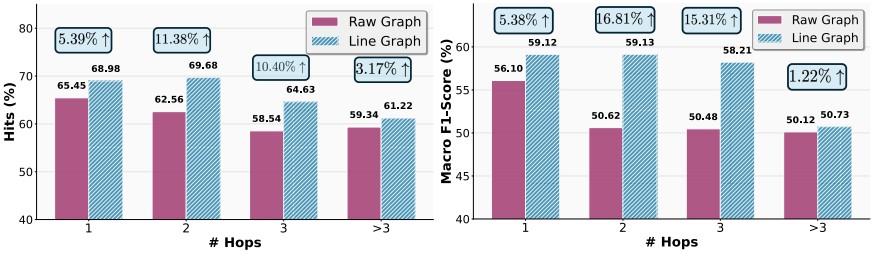

(a) Performance measured by Hits.   (b) Performance measured by Macro-F1.

Figure 2: Performance comparison on CWQ dataset of `REL-RAG` using line-graph and raw-graph representations across questions with different reasoning hops.

CWQ, making it suitable for evaluating cross-dataset generalization when models are trained on WebQSP and tested on GrailQA. In our experiments, we use the pre-extracted subgraphs from Sun et al. (2024b) to curate a dataset, with a knowledge graph for each question. Detailed dataset statistics are provided in Appendix D.

**Baselines.** We compare `REL-RAG` with 15 state-of-the-art baseline methods, encompassing both general LLM without external KGs, and KG-based RAG approaches that integrate KGs with LLM for KGQA. Among them, GNN-RAG and SubgraphRAG utilize graph-based retrievers to extract relevant knowledge from the knowledge graph. For SubgraphRAG, we adopt the same reasoning modules as used in our framework to ensure fair comparisons. We report the performance of SubgraphRAG using the setting with 100 and 500 retrieved triples, as provided in its original paper (Li et al., 2024).

**Evaluation.** Following previous practice, we adopt Hits and Macro-F1 to assess the effectiveness of `REL-RAG`. *Hits* measures whether the correct answer appears among the predictions, while *Macro-F1* computes the average of F1 scores across all test samples.

**Setup.** Following prior work (Li et al., 2024), we employ *gte-large-en-v1.5* (Li et al., 2023b) as the pretrained text encoder which remains frozen throughout training to extract text embeddings to ensure fair comparisons. The graph retriever adopted is a standard GCN without structural modifications. We evaluate both path-based and triple-based training objectives, retrieving either 50 or 500 triples in each configuration. For the reasoning module, we consider GPT-4o, GPT-4o-mini, and instruction-tuned Llama3.1-8B model without fine-tuning.

### 5.2  IN-DISTRIBUTION RESULTS

We first study in-distribution performance of `REL-RAG` in this section. From Table 1, we make three key observations. **(1)** Pure LLMs perform substantially worse than KG+LLM pipelines, underscoring the necessity of knowledge retrieval for KGQA. **(2)** With a 500-triple budget and stronger LLMs, `REL-RAG` delivers the strongest results overall. Under the triple-based objective, `REL-RAG` surpasses the best baselines by up to **3.1%** on WebQSP and **10.3%** on CWQ, demonstrating its ability to exploit richer evidence when the reasoner can handle larger contexts. **(3)** With around 50 triples, the path-based objective proves more precise and token-efficient, making it well-suited to smaller and medium-scale LLMs. For example, when paired with Llama3.1-8B, `REL-RAG` achieves improvements of up to **20.3%** on CWQ in terms of Macro-F1 compared with best baseline method, substantially closing the performance gap between smaller LLMs and stronger ones.

## 5.3 How does line graph transformation affect generalizability?

**Setup.** We keep the model architecture and all training configurations fixed (e.g., learning rate, number of epochs) and only vary the input graph: raw graph versus line graph. Concretely, we train the retriever on WebQSP and evaluate it on CWQ and GrailQA. Since WebQSP consists mainly of 1-hop questions with only 2826 training samples, it provides an ideal testbed for both compositional and OOD generalization. For CWQ, we further break down the evaluation by question hop length (1-hop, 2-hop, 3-hop, and more than 3-hop). For GrailQA, we analyze the top 5 domains with the largest number of questions, which introduces severe compositional and OOD shifts relative to WebQSP. All experiments use triple-based training with 500 retrieved triples, paired with GPT-4o-mini.

**Overall results.** Across all three distribution shifts, `REL-RAG` significantly outperforms its raw-graph counterpart. The gains are particularly pronounced under compositional and OOD shifts, confirming that line graph transformation provides suitable structural bias for generalization.

**A closer look at CWQ.** As shown in Figure 2, `REL-RAG` outperforms the raw-graph counterpart across different hop settings. The improvements are substantial within 3 hops, where Macro-F1 increases by up to $16.81\%$ and Hits by up to $11.38\%$ points. This reflects the advantage of explicitly modeling relation transitions, which eliminates the mixing of entity representation in entity graph.

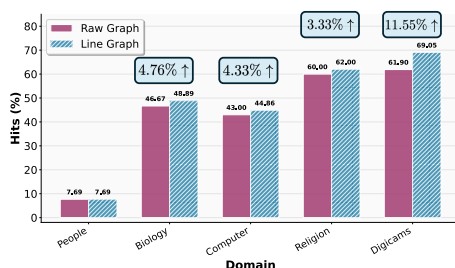

Figure 3: Generalization performance on top 5 domains in GrailQA.

Table 3: Performance comparison of graph retriever architectures using path-based training with 50 retrieved triples. Best results are **bold**; second-best are underlined.

| Graph Retriever | LLM | WebQSP | | CWQ | |
|---|---|---|---|---|---|
| | | Macro-F1 | Hit | Macro-F1 | Hit |
| 1-layer GCN (bi-directional) | Llama3.1-8B | 70.2 | 85.0 | 52.4 | 62.6 |
| | GPT-4o-mini | 74.0 | 90.4 | 51.4 | 62.5 |
| 2-layer GCN (uni-directional) | Llama3.1-8B | 71.6 | 85.5 | 51.4 | 61.1 |
| | GPT-4o-mini | 74.4 | 89.1 | 51.9 | 61.0 |
| 2-layer GCN (bi-directional) | Llama3.1-8B | 77.3 | 88.8 | 56.8 | 67.2 |
| | GPT-4o-mini | **80.4** | **92.5** | **58.1** | **69.3** |

**A closer look at GrailQA.** In GrailQA, many domains differ substantially from those in WebQSP. On the five largest domains, `REL-RAG` surpasses the raw-graph counterpart by an average of $6.04\%$ in Hits, and achieves overall improvements of $5.12\%$ and $5.54\%$ in Hits and Macro-F1 respectively. These results highlight the inductive bias of relation learning can also enable effective generalization across different domains.

## 5.4 Case study

**Case study.** We present an illustrative example where the retriever is trained on WebQSP and evaluated on CWQ. Among the retrieved triples, `REL-RAG` successfully identifies relevant evidence such as 1946 World Series $\xrightarrow{\text{sports.sports\_championship\_event.champion}}$ St. Louis Cardinals and St. Louis Cardinals $\xrightarrow{\text{sports.sports\_team.arena\_stadium}}$ Busch Stadium, which directly link the champion of the 1946 World Series to its home stadium. In contrast, the vanilla KG variant fails to establish this critical connection, highlighting its difficulty in composing multi-hop relations. The comparison is illustrated in Figure 4.

## 5.5 Ablation Study on Retriever Architecture

We analyze how different GNN architectures used in the graph retriever affect KGQA performance. As shown in Table 3, `REL-RAG` with 1-layer GCN with bidirectional message passing is on par with `REL-RAG` with 2-layer uni-directional GCN model. However, when the bidirectional message passing is incorporated into the 2-layer GCN, KGQA performance improves significantly across both datasets and reasoning models. This highlights the importance of the bi-directional message-passing design, which augments the node representations by incorporating information from neighboring nodes from both directions.

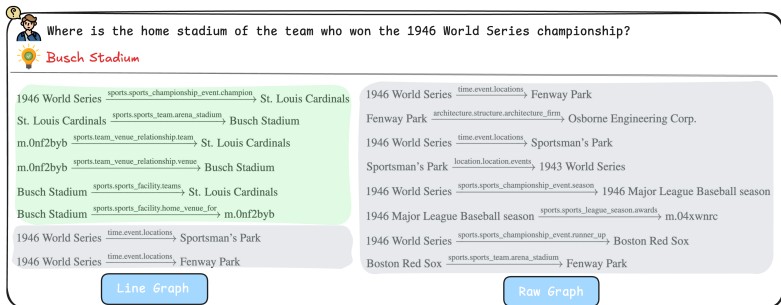

Figure 4: Case study of the triples retrieved by REL-RAG with line-graph and raw-graph representations. Green triples denote evidence relevant to the correct answers, while Gray triples denote irrelevant evidence.

## 6 COMPLEXITY ANALYSIS

**Preprocessing.** Given a graph $\mathcal{G} = (\mathcal{V}, \mathcal{E})$ with $|\mathcal{V}|$ nodes and $|\mathcal{E}|$ edges, The time complexity of this transformation is $\mathcal{O}(|\mathcal{E}|d_{\max})$, where $d_{\max}$ is the maximum node degree in $\mathcal{G}$. The space complexity is $\mathcal{O}(|\mathcal{E}| + |\mathcal{E}'|)$, where $|\mathcal{E}'|$ denotes the number of edges in the resulting line graph $\mathcal{G}'$, typically on the order of $\mathcal{O}(|E|d_{avg})$, with $d_{avg}$ as the average node degree.

**Model training and inference.** For both training and inference, with a $K$-layer GCN operating on the line graph $\mathcal{G}'$, the time complexity is $\mathcal{O}(K|\mathcal{E}'|F)$, where $F$ is the dimensionality of node features. The space complexity is $\mathcal{O}(|\mathcal{V}'|F + |\mathcal{E}'|)$, where $|V'|$ is the number of triples (i.e., nodes in the line graph). During model training and inference, it does not involve any LLM call for the retriever.

## 7 RELATED WORK

**Retrieve-then-reasoning paradigm**. In KG-based RAG, many methods follow a *retrieve-then-reasoning* pipeline (Li et al., 2023a; Kim et al., 2023; Liu et al., 2025; Wu et al., 2023; Wen et al., 2023; Mavromatis & Karypis, 2024; Li et al., 2024; Luo et al., 2024; Mavromatis & Karypis, 2022): a retriever first extracts relevant triples from the KG, and a reasoner generates the final answer from the retrieved evidence. Retrievers are broadly LLM-based or graph-based. LLM-based retrievers offer strong semantic matching but incur hallucination risk and high latency/cost. Lightweight GNN-based retrievers (Mavromatis & Karypis, 2022; Li et al., 2024; Zhang et al., 2022; Mavromatis & Karypis, 2022) operate directly on KG structure and, when paired with LLM reasoners, achieve strong KGQA performance while reducing hallucinations and compute. However, graph-based retrievers still struggle with generalization. We address this issue via REL-RAG, a simple framework that provably improves generalization of graph retrievers, and can flexibly adapt to different token budgets and LLM reasoning capabilities.

**KG-based agentic RAG**. Another line of research leverages LLMs as agents that iteratively explore the knowledge graph to retrieve relevant information (Gao et al., 2024a; Wang et al., 2024; Jiang et al., 2024; Sun et al., 2024a; Chen et al., 2024; Ma et al., 2024; Jin et al., 2024). In this setting, the agent integrates both retrieval and reasoning capabilities, enabling more adaptive knowledge access. While this approach has demonstrated effectiveness in identifying relevant triples, the iterative exploration process incurs latency and computational costs due to repeated LLM calls. In contrast, REL-RAG adopts a graph retriever, which avoids repetitive LLM invocations during knowledge retrieval.

## 8 CONCLUSIONS

In this work, we introduced REL-RAG, a relation-aware RAG framework for KGQA. REL-RAG elevates relations to first-class objects, inducing a structural bias that aligns with relational reasoning in KGs, and provably enhances the generalization ability of graph retrievers. REL-RAG further integrates two complementary learning regimes to flexibly adapt to different token budgets and LLM capacities. Empirical results on KGQA benchmarks demonstrate its state-of-the-art performance, together with strong generalization ability across various types of distribution shift.

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

# Appendix

CONTENTS

## A THEORETICAL ANALYSIS

In this section, we provide detailed proofs for the propositions and theorems in the main paper.

### A.1 PROOF OF PROPOSITION 1

*Proof. Well-definedness.* Consecutive triples in $P$ are $t_i = \langle e_{i-1}, r_i, e_i \rangle$ and $t_{i+1} = \langle e_i, r_{i+1}, e_{i+1} \rangle$. They share the intermediate entity $e_i$ with preserved direction, so there is an edge $t_i \to t_{i+1}$ in $l(\mathcal{G})$ by Definition 4. Hence $\Phi(P) = (t_1 \to t_2 \to \cdots \to t_k)$ is a valid path of length $k-1$ in $l(\mathcal{G})$.

*Injectivity.* If two original paths $P \neq P'$ differ at position $j$ (i.e., $t_j \neq t'_j$), then their images differ at node $j$ in $l(\mathcal{G})$; thus $\Phi(P) \neq \Phi(P')$.

*Surjectivity.* Let $(x_1 \to x_2 \to \cdots \to x_k)$ be any path in $l(\mathcal{G})$, where each $x_i$ corresponds to a triple $t_i = \langle u_{i-1}, s_i, u_i \rangle$ in $\mathcal{G}$. Since $x_i \to x_{i+1}$ is an edge in $l(\mathcal{G})$, the tail of $t_{i+1}$ equals the head of $t_i$, i.e., $u_i = u_i$ is shared. Therefore $(t_1, \ldots, t_k)$ forms a directed path of length $k$ in $\mathcal{G}$, whose image under $\Phi$ is the given path. $\square$

Proposition 1 establishes a one-to-one correspondence between any reasoning path in the original graph $\mathcal{G}$ and a unique path in its line graph $l(\mathcal{G})$, thereby providing the foundation for path-based reasoning on $l(\mathcal{G})$.

## A.2    Proof of Theorem 3.1

To prove Theorem 3.1, we begin with a concise proof sketch, followed by the detailed derivation.

**Proof sketch.** The key insight is that the line graph representation reduces the effective parameter dimension by encoding relation transitions explicitly in the graph structure. We first establish that GCNs operating on the entity graph $\mathcal{G}$ require $O(Rd^2)$ parameters, while GCNs on the line graph $\mathcal{G}'$ require only $O(d^2)$ parameters when $R \leq d$. Using covering number arguments and Dudley's entropy integral theorem, we derive Rademacher complexity bounds for the entity graph and line graph. Standard generalization theory then translates these into the stated bounds.

### A.2.1    Functional Forms of GCN Models on Different Graph Representations

We first characterize the parameter spaces for GCNs operating on the entity graph versus the line graph.

**Entity graph.** Let $\mathcal{R}$ be the set of relations with $R := |\mathcal{R}|$, and let $E$ be the set of edges. Let $h_u^{(0)} \in \mathbb{R}^d$ be the input embedding of entity $u$ and $\{W_r \in \mathbb{R}^{d \times d} : r \in \mathcal{R}\}$ the relation-specific weight matrices. At a node $v$, the first-hop update aggregates across all relations:

$$h_v^{(1)} = \sigma \left( \sum_{r \in \mathcal{R}} \sum_{u:(u,r,v) \in E} W_r h_u^{(0)} \right), \tag{16}$$

and at node $w$ the second hop is:

$$h_w^{(2)} = \sigma \left( \sum_{r \in \mathcal{R}} \sum_{v:(v,r,w) \in E} W_r h_v^{(1)} \right). \tag{17}$$

Note that even when evaluating a specific two-hop path $(u, r_1, v, r_2, w)$, the intermediate state $h_v^{(1)}$ mixes messages from *all* incident relations at $v$ (not only $r_1$), and $h_w^{(2)}$ again mixes all relations incident to $w$.

We score $(u, w)$ by projecting onto a bounded test vector $\phi_w \in \mathbb{R}^d$ with $\|\phi_w\|_2 \leq 1$:

$$f_{\mathbf{W}}(u, w) = \langle h_w^{(2)}, \phi_w \rangle. \tag{18}$$

Define the hypothesis class:

$$\mathcal{H}_{\mathcal{G}} = \{f_{\mathbf{W}}(\cdot, \cdot) : \|W_r\|_F \leq B \text{ for all } r \in \mathcal{R}\}. \tag{19}$$

The parameter space is $\mathcal{W}_{\mathcal{G}} := \prod_{r=1}^{R} \{W_r \in \mathbb{R}^{d \times d} : \|W_r\|_F \leq B\}$. Vectorizing and stacking gives:

$$\text{vec}(\mathbf{W}) = \left[ \text{vec}(W_1)^\top, \text{vec}(W_2)^\top, \ldots, \text{vec}(W_R)^\top \right]^\top \in \mathbb{R}^{Rd^2}. \tag{20}$$

Thus, the parameter space for the entity graph model is $O(Rd^2)$.

**Line graph.** In the line graph representation, each node corresponds to a triple $(h, r, t)$. A two-hop query $(u, r_1, r_2, w)$ becomes two adjacent line graph nodes: $(u, r_1, v)$ and $(v, r_2, w)$. A GCN on the line graph directly models relation-to-relation transitions:

$$h_{\mathcal{G}'}(r_1, r_2) = \phi_{r_1}^\top M \phi_{r_2}, \tag{21}$$

where $M \in \mathbb{R}^{d \times d}$ is a shared transformation matrix and $\{\phi_r \in \mathbb{R}^d : r \in \mathcal{R}\}$ are relation embeddings with $\|\phi_r\|_2 \leq 1$.

Define the hypothesis class:

$$\mathcal{H}_{\mathcal{G}'} = \{h(r_1, r_2) = \langle \phi_{r_1}, M\phi_{r_2} \rangle : \|M\|_F \leq B, \|\phi_r\|_2 \leq 1 \text{ for all } r\}. \tag{22}$$

The parameter space consists of $M \in \mathbb{R}^{d \times d}$ and $R$ relation embeddings in $\mathbb{R}^d$, totaling $d^2 + Rd$ parameters. Under the assumption that $R \leq d$ (which holds in most practical knowledge graphs), the parameter space is $O(d^2)$.

### A.2.2 Formal Proof via Covering Numbers

We establish the necessary definitions and apply Dudley's entropy integral theorem.

**Definition ($\varepsilon$-covering).** For a metric space $(X, d)$ and a subset $S \subseteq X$, an $\varepsilon$-covering of $S$ is a finite set $C_\varepsilon$ such that for every $s \in S$, there exists $c \in C_\varepsilon$ with $d(s, c) \leq \varepsilon$. The $\varepsilon$-covering number $\mathcal{N}(\varepsilon, S, d)$ is the cardinality of the smallest $\varepsilon$-covering of $S$.

**Definition (Empirical Rademacher Complexity).** For a hypothesis class $\mathcal{H}$ and a sample $S = \{x_1, \ldots, x_m\}$, the empirical Rademacher complexity is:

$$\hat{\mathfrak{R}}_S(\mathcal{H}) = \mathbb{E}_\sigma \left[ \sup_{h \in \mathcal{H}} \frac{1}{m} \sum_{i=1}^{m} \sigma_i h(x_i) \right], \tag{23}$$

where $\sigma_i$ are independent Rademacher random variables taking values in $\{-1, +1\}$ with equal probability.

**Theorem (Dudley's Entropy Integral (Dudley, 1967)).** For a hypothesis class $\mathcal{H}$ with bounded functions $|h(x)| \leq 1$, the Rademacher complexity can be bounded as:

$$\mathfrak{R}_m(\mathcal{H}) \leq \inf_{\alpha > 0} \left( 4\alpha + \frac{12}{\sqrt{m}} \int_\alpha^1 \sqrt{\log \mathcal{N}(\varepsilon, \mathcal{H}, \|\cdot\|_\infty)} d\varepsilon \right). \tag{24}$$

**Step 1: Covering numbers for parameter spaces.**

For the entity graph model with parameter space $\mathcal{W}_{\mathcal{G}} \subset \mathbb{R}^{Rd^2}$, by standard volumetric arguments for Euclidean balls:

$$\log \mathcal{N}(\varepsilon, \mathcal{W}_{\mathcal{G}}, \|\cdot\|_2) \leq Rd^2 \log \left( \frac{cB}{\varepsilon} \right), \tag{25}$$

where $c > 0$ is an absolute constant.

For the line graph model with parameter space $\mathcal{W}_{\mathcal{G}'} \subset \mathbb{R}^{d^2 + Rd}$, assuming $R \leq d$:

$$\log \mathcal{N}(\varepsilon, \mathcal{W}_{\mathcal{G}'}, \|\cdot\|_2) \leq 2d^2 \log \left( \frac{cB}{\varepsilon} \right) = O \left( d^2 \log \frac{B}{\varepsilon} \right). \tag{26}$$

**Step 2: Applying Dudley's theorem.**

Setting $\alpha = 1/\sqrt{m}$ and evaluating the integral:

For the entity graph model:

$$\mathfrak{R}_m(\mathcal{H}_{\mathcal{G}}) \leq \frac{C}{\sqrt{m}} \int_{1/\sqrt{m}}^1 \sqrt{Rd^2 \log \frac{B}{\varepsilon}} \, d\varepsilon \tag{27}$$

$$= \frac{C\sqrt{Rd^2}}{\sqrt{m}} \int_{1/\sqrt{m}}^1 \sqrt{\log \frac{B}{\varepsilon}} \, d\varepsilon \tag{28}$$

$$= O \left( \frac{d\sqrt{R}\sqrt{\log m}}{\sqrt{m}} \right) = O \left( \frac{\sqrt{R}d}{\sqrt{m}} \right). \tag{29}$$

For the line graph model:

$$\mathfrak{R}_m(\mathcal{H}_{\mathcal{G}'}) \leq \frac{C}{\sqrt{m}} \int_{1/\sqrt{m}}^1 \sqrt{d^2 \log \frac{B}{\varepsilon}} \, d\varepsilon \tag{30}$$

$$= \frac{Cd}{\sqrt{m}} \int_{1/\sqrt{m}}^1 \sqrt{\log \frac{B}{\varepsilon}} \, d\varepsilon \tag{31}$$

$$= O\left(\frac{d\sqrt{\log m}}{\sqrt{m}}\right) = O\left(\frac{d}{\sqrt{m}}\right). \tag{32}$$

**Step 3: Generalization bounds.**

Applying the standard Rademacher-based generalization theorem (Bartlett & Mendelson, 2003), with probability at least $1 - \delta$ over an i.i.d. sample of size $m$:

$$\mathcal{L}(h) \leq \hat{\mathcal{L}}_m(h) + 2\mathfrak{R}_m(\mathcal{H}) + \sqrt{\frac{\log(1/\delta)}{2m}}. \tag{33}$$

Therefore, for the entity graph model:

$$\boxed{\mathcal{L}(h_{\mathcal{G}}) \leq \hat{\mathcal{L}}_m(h_{\mathcal{G}}) + O\left(\frac{\sqrt{R}d}{\sqrt{m}}\right) + \sqrt{\frac{\log(1/\delta)}{2m}}} \tag{34}$$

And for the line graph model:

$$\boxed{\mathcal{L}(h_{\mathcal{G}'}) \leq \hat{\mathcal{L}}_m(h_{\mathcal{G}'}) + O\left(\frac{d}{\sqrt{m}}\right) + \sqrt{\frac{\log(1/\delta)}{2m}}} \tag{35}$$

The $O(\sqrt{R})$ improvement factor for line graph models directly translates to tighter generalization guarantees. This advantage arises because relation transitions are encoded explicitly in the graph structure, eliminating the need for per-node mixing and demixing of relation-specific messages required in the entity graph formulation. $\qquad\square$

## A.3 PROOF OF THEOREM 3.2

**Proof sketch.** We extend the PAC-Bayes bound by decomposing $\mathcal{L}_{\mathcal{D}_T}(Q) = \mathcal{L}_{\mathcal{D}_S}(Q) + [\mathcal{L}_{\mathcal{D}_T}(Q) - \mathcal{L}_{\mathcal{D}_S}(Q)]$, where the shift term is bounded by $\epsilon_{\text{shift}}$. For line graphs, decomposing $|h_{\mathcal{G}'} - h_{\mathcal{G}'}^*|$ yields $O(\sqrt{d/\min\{N(r_1), N(r_2)\}})$. For entity graphs, Lipschitz continuity over $R$ weight matrices introduces a $\sqrt{R}$ factor, giving $O(\sqrt{Rd/\min\{N(r_1), N(r_2)\}})$.

**Theorem A.1** (PAC-Bayes). *Let $P$ be any prior over a hypothesis class $\mathcal{H}$. For any posterior $Q$ chosen after observing an i.i.d. sample $S = \{(x_i, y_i)\}_{i=1}^m \sim \mathcal{D}^m$, with probability at least $1 - \delta$,*

$$\mathcal{L}_{\mathcal{D}}(Q) \leq \mathcal{L}_S(Q) + \sqrt{\frac{KL(Q\|P) + \log(2\sqrt{m}/\delta)}{2m}}. \tag{36}$$

Theorem A.1 can be directly obtained from (McAllester, 1998). For training and test distributions $\mathcal{D}_S, \mathcal{D}_T$, define:

$$\epsilon_{\text{shift}}(\mathcal{D}_S, \mathcal{D}_T) := \sup_{h \in \mathcal{H}} |\mathcal{L}_{\mathcal{D}_T}(h) - \mathcal{L}_{\mathcal{D}_S}(h)|. \tag{37}$$

Expanding Theorem A.1 yields:

$$\mathcal{L}_{\mathcal{D}_T}(Q) = \mathcal{L}_{\mathcal{D}_S}(Q) + \left[\mathcal{L}_{\mathcal{D}_T}(Q) - \mathcal{L}_{\mathcal{D}_S}(Q)\right], \tag{38}$$

and

$$|\mathcal{L}_{\mathcal{D}_T}(Q) - \mathcal{L}_{\mathcal{D}_S}(Q)| = \left|\mathbb{E}_{h \sim Q}\left[\mathcal{L}_{\mathcal{D}_T}(h) - \mathcal{L}_{\mathcal{D}_S}(h)\right]\right| \leq \epsilon_{\text{shift}}(\mathcal{D}_S, \mathcal{D}_T). \tag{39}$$

Combined with Theorem A.1, we obtain Eqn. equation 7 in Theorem 3.2.

**Line graph.** Consider the line graph scorer for a two-hop composition $(r_1, r_2)$:

$$h_{\mathcal{G}'}(r_1 \circ r_2) = \langle \phi_{r_1}, W\phi_{r_2} \rangle, \qquad \phi_r \in \mathbb{R}^d, \ W \in \mathbb{R}^{d \times d}. \tag{40}$$

Let $h_{\mathcal{G}'}^*(r_1 \circ r_2) = \langle \phi_{r_1}^*, W^*\phi_{r_2}^* \rangle$ denote the ground-truth model. Assume spectral norm bounds $\|W\|_2, \|W^*\|_2 \leq B$ and embedding bounds $\|\phi_r\|_2, \|\phi_r^*\|_2 \leq 1$.

With sub-Gaussian concentration, learning $\phi_r$ from $N(r)$ sub-Gaussian samples implies:

$$\|\phi_r - \phi_r^*\|_2 \leq C\sqrt{\frac{d + \log(1/\eta)}{N(r)}} \quad \text{with probability at least } 1 - \eta, \tag{41}$$

where $C > 0$ absorbs the sub-Gaussian constant.

Similarly, for the shared matrix $W$ learned from all training samples:

$$\|W - W^*\|_F \leq C'\sqrt{\frac{d^2 + \log(1/\eta)}{m}} \quad \text{with probability at least } 1 - \eta. \tag{42}$$

Now we decompose the error:

$$
\begin{aligned}
\left| h_{\mathcal{G}'} - h_{\mathcal{G}'}^* \right| &= \left| \langle \phi_{r_1}, W\phi_{r_2} \rangle - \langle \phi_{r_1}^*, W^*\phi_{r_2}^* \rangle \right| \\
&\leq \underbrace{\left| \langle \phi_{r_1} - \phi_{r_1}^*, W\phi_{r_2} \rangle \right|}_{(A)} + \underbrace{\left| \langle \phi_{r_1}^*, W(\phi_{r_2} - \phi_{r_2}^*) \rangle \right|}_{(B)} \\
&\quad + \underbrace{\left| \langle \phi_{r_1}^*, (W - W^*)\phi_{r_2}^* \rangle \right|}_{(C)}.
\end{aligned}
\tag{43}
$$

Bounding each term:

$$(A) = \left| \langle \phi_{r_1} - \phi_{r_1}^*, W\phi_{r_2} \rangle \right| \leq \|W\|_2 \|\phi_{r_1} - \phi_{r_1}^*\|_2 \|\phi_{r_2}\|_2 \tag{44}$$

$$\leq BC\sqrt{\frac{d + \log(1/\eta)}{N(r_1)}} = O\left( \sqrt{\frac{d + \log(1/\eta)}{N(r_1)}} \right), \tag{45}$$

and symmetrically:

$$(B) = O\left( \sqrt{\frac{d + \log(1/\eta)}{N(r_2)}} \right). \tag{46}$$

For term (C), when $m \geq \min\{N(r_1), N(r_2)\}$:

$$(C) = \left| \langle \phi_{r_1}^*, (W - W^*)\phi_{r_2}^* \rangle \right| \leq \|\phi_{r_1}^*\|_2 \|W - W^*\|_F \|\phi_{r_2}^*\|_2 \tag{47}$$

$$\leq C'\sqrt{\frac{d^2 + \log(1/\eta)}{m}} = O\left( \sqrt{\frac{d}{m}} \right) = O\left( \sqrt{\frac{d}{\min\{N(r_1), N(r_2)\}}} \right). \tag{48}$$

Therefore:

$$\left| h_{\mathcal{G}'}(r_1 \circ r_2) - h_{\mathcal{G}'}^*(r_1 \circ r_2) \right| \leq C_1 \left( \sqrt{\frac{d + \log(1/\eta)}{N(r_1)}} + \sqrt{\frac{d + \log(1/\eta)}{N(r_2)}} \right). \tag{49}$$

Using $\sqrt{a} + \sqrt{b} \leq 2\sqrt{\max\{a, b\}}$:

$$\left| h_{\mathcal{G}'}(r_1 \circ r_2) - h_{\mathcal{G}'}^*(r_1 \circ r_2) \right| \leq 2C_1 \sqrt{\frac{d + \log(1/\eta)}{\min\{N(r_1), N(r_2)\}}} \quad \text{w.p.} \ \geq 1 - 3\eta. \tag{50}$$

Taking expectation over the randomness in training, and setting $\eta = 1/m$, we obtain:

$$\boxed{\epsilon_{\text{shift}}^{\mathcal{G}'} = O\left( \sqrt{\frac{d}{\min\{N(r_1), N(r_2)\}}} \right).} \tag{51}$$

**Entity graph.** Recall the two-layer aggregation over relations:

$$h_v^{(1)} = \sigma \left( \sum_{r \in \mathcal{R}} \sum_{u:(u,r,v) \in E} W_r h_u^{(0)} \right), \quad h_w^{(2)} = \sigma \left( \sum_{r \in \mathcal{R}} \sum_{v:(v,r,w) \in E} W_r h_v^{(1)} \right), \qquad (52)$$

and the score $f_{\mathbf{W}}(u,w) = \langle h_w^{(2)}, \phi_w \rangle$ with $\|\phi_w\|_2 \le 1$.

Let $\mathbf{W}^* = (W_r^*)_{r \in \mathcal{R}}$ denote the ground truth and $\Delta_r := W_r - W_r^*$ the estimation error.

**Per-relation estimation rates.** Assume each relation $r$ participates in $N(r)$ i.i.d. sub-Gaussian training instances (either as first or second hop), with $\|W_r^*\|_2 \le B$ and $\|\phi_e\|_2 \le 1$. For structured matrices (e.g., low-rank or sparse, which is common in knowledge graphs), standard matrix estimation yields (w.p. $\ge 1 - \eta$):

$$\|\Delta_r\|_F \le C_2 \sqrt{\frac{d + \log(1/\eta)}{N(r)}}. \qquad (53)$$

Assume there exists a constant $L > 0$ such that for any parameter tuples $\mathbf{W} = (W_r)_{r \in \mathcal{R}}$ and $\mathbf{W}^* = (W_r^*)_{r \in \mathcal{R}}$ and any two-hop query $(u, w)$:

$$\left| f_{\mathbf{W}}(u,w) - f_{\mathbf{W}^*}(u,w) \right| \le L \sum_{r \in \mathcal{R}} \|W_r - W_r^*\|_F. \qquad (54)$$

By Cauchy-Schwarz, this also implies:

$$\left| f_{\mathbf{W}}(u,w) - f_{\mathbf{W}^*}(u,w) \right| \le L \sqrt{R} \left( \sum_{r \in \mathcal{R}} \|W_r - W_r^*\|_F^2 \right)^{1/2}. \qquad (55)$$

Let $\Delta_r := W_r - W_r^*$. Using the per-relation estimation rate $\|\Delta_r\|_F \le C_2 \sqrt{(d + \log(1/\eta))/N(r)}$ (with probability at least $1 - \eta$), and noting that all $R$ relations contribute to the error:

$$\left| h_{\mathcal{G}}(r_1 \circ r_2) - h_{\mathcal{G}}^*(r_1 \circ r_2) \right| \le L \sum_{r \in \mathcal{R}} \|\Delta_r\|_F \qquad (56)$$

$$\le L \sqrt{R} \left( \|\Delta_{r_1}\|_F^2 + \|\Delta_{r_2}\|_F^2 \right)^{1/2} \qquad (57)$$

$$\le L \sqrt{R} \left( \|\Delta_{r_1}\|_F + \|\Delta_{r_2}\|_F \right) \qquad (58)$$

$$\le L C_2 \sqrt{R} \left( \sqrt{\frac{d + \log(1/\eta)}{N(r_1)}} + \sqrt{\frac{d + \log(1/\eta)}{N(r_2)}} \right). \qquad (59)$$

Using $\sqrt{a} + \sqrt{b} \le 2\sqrt{\max\{a,b\}}$:

$$\left| h_{\mathcal{G}}(r_1 \circ r_2) - h_{\mathcal{G}}^*(r_1 \circ r_2) \right| \le 2 L C_2 \sqrt{\frac{R(d + \log(1/\eta))}{\min\{N(r_1), N(r_2)\}}} \quad \text{w.p. } \ge 1 - \eta. \qquad (60)$$

As before, taking expectation over the randomness (setting $\eta = 1/m$):

$$\boxed{\epsilon_{\text{shift}}^{\mathcal{G}} = O \left( \sqrt{\frac{Rd}{\min\{N(r_1), N(r_2)\}}} \right).} \qquad (61)$$

In summary, the entity graph model carries an extra $\sqrt{R}$ factor because each node aggregates messages over all $R$ incident relations at every hop, so prediction sensitivity to parameter error scales as $\sqrt{R}$. By contrast, the line graph encodes relation transitions explicitly and avoids per-node mixing, yielding uniformly tighter compositional generalization bounds. $\qquad \square$

### A.4 PROOF OF THEOREM 3.3

Fix an unseen relation $r_{\text{new}} \notin \mathcal{R}_{\text{train}}$ and let

$$\Delta = \min_{r \in \mathcal{R}_{\text{train}}} d(r_{\text{new}}, r), \qquad r_{\text{sim}} = \arg\min_{r \in \mathcal{R}_{\text{train}}} d(r_{\text{new}}, r). \tag{62}$$

We work under the same confidence event as Theorem 3.2 so that the PAC-Bayesian inequality equation 7 holds; it suffices to upper bound the distribution-shift term $\epsilon_{\text{shift}}^{l(\mathcal{G})}$ for the line-graph model and to lower bound $\epsilon_{\text{shift}}^{\mathcal{G}}$ for the entity-graph model.

**Line graph.** Recall the line-graph predictor $h_{\mathcal{G}'}$ scores a two-hop composition via the bilinear form:

$$h_{\mathcal{G}'}(r_1 \circ r_2) = \langle \phi_{r_1}, W \phi_{r_2} \rangle, \qquad \phi_r \in \mathbb{R}^d, \ W \in \mathbb{R}^{d \times d}. \tag{63}$$

We assume: (i) the loss $\ell(\cdot, y)$ is 1-Lipschitz in its first argument for all labels $y$; (ii) the predictor $h_{\mathcal{G}'}$ is $L$-Lipschitz w.r.t. the relation representation (in the metric $d(\cdot, \cdot)$ on relations); (iii) embeddings are bounded $\|\phi_r\|_2 \le 1$, and $\|W\|_2 \le B$ (these constants only scale multiplicative factors).

Consider any composition where $r_{\text{new}}$ appears (symmetrically as the first or the second hop). For definiteness, take pairs of the form $(r_{\text{new}}, r')$ with $r'$ seen. For a single example $(u, w, r_{\text{new}}, r', y)$ we decompose the loss difference by inserting $r_{\text{sim}}$:

$$\left| \ell\big(h_{\mathcal{G}'}(r_{\text{new}}, r'), y\big) - \ell\big(h_{\mathcal{G}'}^*(r_{\text{new}}, r'), y\big) \right| \le \underbrace{\left| \ell\big(h_{\mathcal{G}'}(r_{\text{new}}, r'), y\big) - \ell\big(h_{\mathcal{G}'}(r_{\text{sim}}, r'), y\big) \right|}_{(A)}$$

$$+ \underbrace{\left| \ell\big(h_{\mathcal{G}'}(r_{\text{sim}}, r'), y\big) - \ell\big(h_{\mathcal{G}'}^*(r_{\text{sim}}, r'), y\big) \right|}_{(B)} \tag{64}$$

$$+ \underbrace{\left| \ell\big(h_{\mathcal{G}'}^*(r_{\text{sim}}, r'), y\big) - \ell\big(h_{\mathcal{G}'}^*(r_{\text{new}}, r'), y\big) \right|}_{(C)}.$$

Here $h_{\mathcal{G}'}^*$ denotes the Bayes (ground-truth) score in the same bilinear form.

*Semantic terms* (A) *and* (C). By the 1-Lipschitz property of $\ell$ and the $L$-Lipschitz property of $h_{\mathcal{G}'}$ with respect to relation arguments,

$$(A) \le \left| h_{\mathcal{G}'}(r_{\text{new}}, r') - h_{\mathcal{G}'}(r_{\text{sim}}, r') \right| \le L\, d(r_{\text{new}}, r_{\text{sim}}) = L\,\Delta. \tag{65}$$

An identical argument gives $(C) \le L\,\Delta$. We absorb these two semantic terms into a single $O(L\,\Delta)$ contribution.

*Estimation term* (B). By the line-graph estimation analysis (as in the compositional shift proof), learning a relation embedding from $N(r)$ i.i.d. sub-Gaussian samples yields the concentration rate

$$\|\phi_r - \phi_r^*\|_2 = O\Big(\sqrt{d/N(r)}\Big) \quad \text{w.h.p.} \tag{66}$$

Using Cauchy–Schwarz and $\|W\|_2 \le B$, $\|\phi_{r'}\|_2 \le 1$, we obtain for the score error

$$\left| h_{\mathcal{G}'}(r_{\text{sim}}, r') - h_{\mathcal{G}'}^*(r_{\text{sim}}, r') \right| \le B \|\phi_{r_{\text{sim}}} - \phi_{r_{\text{sim}}}^*\|_2 \|\phi_{r'}\|_2 = O\Big(\sqrt{d/N(r_{\text{sim}})}\Big), \tag{67}$$

and hence, by the 1-Lipschitz property of $\ell$,

$$(B) = O\Big(\sqrt{d/N(r_{\text{sim}})}\Big). \tag{68}$$

Combining the three pieces in equation 64 and taking expectations over examples where $r_{\text{new}}$ appears (and finally a supremum over $h$ in the line-graph class), we obtain

$$\boxed{\epsilon_{\text{shift}}^{l(\mathcal{G})} = O\left( L\,\Delta + \sqrt{\frac{d}{N(r_{\text{sim}})}} \right).} \tag{69}$$

The same bound holds when $r_{\text{new}}$ appears in the second hop by symmetry.

**Entity graph.** Consider a two-hop query $(u, r_{\text{new}}, v, r_2, w)$ where $r_{\text{new}} \notin \mathcal{R}_{\text{train}}$. The entity-graph model computes

$$f_{\mathbf{W}}(u, w; r_{\text{new}}, r_2) = \langle h_w^{(2)}, \phi_w \rangle,$$

$$h_w^{(2)} = \sigma\Big( \sum_{r \in \mathcal{R}} \sum_{v':(v',r,w) \in E} W_r h_{v'}^{(1)} \Big), \tag{70}$$

with $\|\phi_w\|_2 \leq 1$. Since $r_{\text{new}}$ is unseen, $W_{r_{\text{new}}}$ remains at random initialization. Let $r_{\text{sim}} = \arg\min_{r \in \mathcal{R}_{\text{train}}} d(r_{\text{new}}, r)$ as before. Define $\hat{\mathbf{W}}$ as the learned parameters where $W_{r_{\text{new}}}$ is replaced by $W_{r_{\text{sim}}}$, and let $\mathbf{W}^*$ denote the ground truth.

For a 1-Lipschitz loss $\ell$ in its first argument,

$$\big| \ell(f_{\mathbf{W}}(u, w; r_{\text{new}}, r_2), y) - \ell(f_{\mathbf{W}^*}(u, w; r_{\text{new}}, r_2), y) \big|$$

$$\leq \underbrace{\big| \ell(f_{\mathbf{W}}(u, w; r_{\text{new}}, r_2), y) - \ell\big(f_{\hat{\mathbf{W}}}(u, w; r_{\text{sim}}, r_2), y\big) \big|}_{(A)} + \underbrace{\big| \ell\big(f_{\hat{\mathbf{W}}}(u, w; r_{\text{sim}}, r_2), y\big) - \ell(f_{\mathbf{W}^*}(u, w; r_{\text{sim}}, r_2), y) \big|}_{(B)}$$

$$+ \underbrace{\big| \ell(f_{\mathbf{W}^*}(u, w; r_{\text{sim}}, r_2), y) - \ell(f_{\mathbf{W}^*}(u, w; r_{\text{new}}, r_2), y) \big|}_{(C)}. \tag{71}$$

*Semantic terms* (A) *and* (C). Assume the scorer is $L$-Lipschitz in the relation argument w.r.t. $d(\cdot, \cdot)$ (holding parameters fixed):

$$(A) \leq L \cdot \big| f_{\mathbf{W}}(u, w; r_{\text{new}}, r_2) - f_{\hat{\mathbf{W}}}(u, w; r_{\text{sim}}, r_2) \big| \leq L \cdot \Delta. \tag{72}$$

By the same $L$-Lipschitzness in the relation argument for the ground truth,

$$(C) \leq L \cdot \Delta. \tag{73}$$

*Estimation term* (B). Adopt the simple parameter Lipschitz condition: there exists $L > 0$ such that

$$\big| f_{\hat{\mathbf{W}}}(u, w; r_{\text{sim}}, r_2) - f_{\mathbf{W}^*}(u, w; r_{\text{sim}}, r_2) \big| \leq L \sum_{r \in \mathcal{R}} \|W_r - W_r^*\|_F \leq L\sqrt{R} \Big( \sum_r \|W_r - W_r^*\|_F^2 \Big)^{1/2}. \tag{74}$$

If relation $r$ is trained with $N(r)$ sub-Gaussian samples, standard concentration yields (with probability $\geq 1 - \eta$)

$$\|W_r - W_r^*\|_F \leq C \sqrt{\frac{d + \log(1/\eta)}{N(r)}}. \tag{75}$$

Retaining the dominant contribution for the path's relation $r_{\text{sim}}$ (others contribute similarly but do not change the scaling),

$$(B) = O\left( \sqrt{\frac{R\,d}{N(r_{\text{sim}})}} \right). \tag{76}$$

Taking expectation over test examples involving $r_{\text{new}}$ and the supremum over $\mathcal{H}_G$,

$$\boxed{\epsilon_{\text{shift}}^{\mathcal{G}} = \sup_{h \in \mathcal{H}_G} \big| \mathcal{L}_{\mathcal{D}_T}(h) - \mathcal{L}_{\mathcal{D}_S}(h) \big| = O\left( L \cdot \Delta + \sqrt{\frac{R\,d}{N(r_{\text{sim}})}} \right).} \tag{77}$$

Similarly, line graph also benefits from the reduced factor $\sqrt{R}$, however, both data representations rely on the semantic similarity to reduce the distribution gap in out-of-domain scenarios.

# B  ADDITIONAL DETAILS ON GRAPH RETRIEVER ARCHITECTURE IN REL−RAG

**Retriever architecture in REL−RAG.** We adopt a 2-layer GCN as the base retriever. Since $\mathcal{G}_q'$ is a directed graph, node representations $\mathbf{z}_i$ would only aggregate information from predecessors $v_{q(0)}, \ldots, v_{q(i-1)}$, which can be suboptimal for predicting the next action. To overcome this limitation,

we employ *bidirectional message passing*: two GCNs are maintained, one operating on $\mathcal{G}'_q$ and the other on its edge-reversed counterpart $\overleftarrow{\mathcal{G}}'_q$. The final representation of each node is obtained by averaging the forward and backward embeddings, thereby incorporating context from both incoming and outgoing neighbors.

$$\overrightarrow{\mathbf{z}}_i = f_{\overrightarrow{\theta}}\left(v_i; \mathcal{G}'_q\right),$$

$$\overleftarrow{\mathbf{z}}_i = f_{\overleftarrow{\theta}}\left(v_i; \overleftarrow{\mathcal{G}}'_q\right), \tag{78}$$

$$\mathbf{z}_i = \text{MEAN}\left(\overrightarrow{\mathbf{z}}_i, \overleftarrow{\mathbf{z}}_i\right).$$

This design allows each node to incorporate contextual signals from both its predecessors and successors, alleviating the limitation of strictly forward-only propagation.

**Inference.** At inference time, we adopt different procedures depending on the training objective:

*Path-based inference.* We first sample the initial question triple $\widetilde{v}_{q(0)}$ as the starting point, then iteratively expand reasoning steps by predicting the next triple (neighbors of previous step) at each step according to the probability score obtained from the softmax loss in Eq. 11, when the *stop* node is sampled or a *max_depth* is reached, the sampling procedure terminates. This process continues until the specified retrieval budget is reached (e.g., 500 triples).

*Triple-based inference.* During inference, triple-based learning directly ranks all triples in $\mathcal{G}'_q$ by their scores $\langle \mathbf{z}_q, \mathbf{z}_v \rangle$ and retrieves the top-$k$ triples.

**Acquiring training labels.** A widely adopted strategy for obtaining training labels is to use the shortest path between the question entity and the answer entity. However, this approach can introduce noise: while some shortest paths are indeed rational and align with human reasoning, many others are spurious, exploiting incidental graph connections that happen to reach the answer but provide little explanatory value. We also include 3 examples in Appendix to illustrate it.

In `REL-RAG`, after collecting shortest paths, we employ an LLM to filter and select the most relevant ones as training signals. This refinement reduces noise and yields more faithful supervision. We observe that such LLM-augmented labels are helpful when the reasoning model is constrained by limited token budgets, since more compact and rational paths facilitate efficient inference. However, when paired with stronger LLM reasoners, the performance difference between LLM-annotated labels and plain shortest-path labels diminishes. One explanation is that, although shortest-path labels contain noise, they still include rational signals within the retrieved triples. Powerful LLMs, given a larger retrieval budget, can effectively identify and leverage these rational cues, thereby narrowing the gap between the two labeling strategies. We provide ablation studies on the training labels in Appendix F.

## C  ALGORITHMIC PSEUDOCODE

We provide the pseudo-code for `REL-RAG` in this section, as shown in Algorithm 1.

## D  DATASETS

WebQSP is a benchmark dataset for KGQA, derived from the original WebQuestions dataset (Berant et al., 2013). It comprises 4,737 natural language questions annotated with full semantic parses in the form of SPARQL queries executable against Freebase. The dataset emphasizes single-hop questions, typically involving a direct relation between the question and answer entities.

CWQ dataset extends the WebQSP dataset to address more challenging multi-hop question answering scenarios. It contains 34,689 complex questions that require reasoning over multiple facts and relations. Each question is paired with a SPARQL query and corresponding answers, facilitating evaluation in both semantic parsing and information retrieval contexts. The datasets statistics can be found in Table 4.

GrailQA is a large-scale KGQA benchmark introduced in (Gu et al., 2021) to evaluate different levels of generalization. Unlike WebQSP and CWQ, the original GrailQA release does not provide

---

**Algorithm 1** Training `REL-RAG` with Line-Graph Transformation and Bidirectional Message Passing

---

**Require:** Training set $\mathcal{D} = \{(q, e_q, e_a, \mathcal{G}_q)\}$; supervision for each $q$ objective selector $o \in \{\text{PATH}, \text{TRIPLE}\}$; epochs $E$; learning rate $\eta$.

**Ensure:** Optimized retriever parameters $\theta = \{\overrightarrow{\theta}, \overleftarrow{\theta}\}$.

    **Initialization**

1: Initialize two GCN encoders $f_{\overrightarrow{\theta}}$ and $f_{\overleftarrow{\theta}}$.

    **Training**

2: **for** $e = 1$ **to** $E$ **do**

3:     **for all** minibatch $\mathcal{B} \subset \mathcal{D}$ **do**

4:         **for all** $(q, e_q, e_a, \mathcal{G}_q) \in \mathcal{B}$ **do**

5:             **Line-graph transform:** $\mathcal{G}'_q \leftarrow \text{LINEGRAPH}(\mathcal{G}_q)$

6:             **Bidirectional embeddings** (Eq. 78):

$$\overrightarrow{\mathbf{z}}_i = f_{\overrightarrow{\theta}}(v_i; \mathcal{G}'_q), \quad \overleftarrow{\mathbf{z}}_i = f_{\overleftarrow{\theta}}(v_i; \overleftarrow{\mathcal{G}'_q}), \quad \mathbf{z}_i = \text{MEAN}\left(\overrightarrow{\mathbf{z}}_i, \overleftarrow{\mathbf{z}}_i\right)$$

7:             **if** $o = \text{PATH}$ **then**

8:                 Compute $\mathcal{L}_{\text{path}}(q; \theta)$ via log-softmax over next-step candidates (Eq. 11)

9:             **else if** $o = \text{TRIPLE}$ **then**

10:                 Form $(\mathcal{V}_{pos}, \mathcal{V}_{neg})$ for $q$; compute $\mathcal{L}_{\text{triple}}(q; \theta)$ (Eq. 14)

11:             **end if**

12:         **end for**

13:         **Minimize loss** by Adam: $\theta \leftarrow \theta - \eta \nabla_\theta \left(\sum_{q \in \mathcal{B}} \mathcal{L}_o(q; \theta)\right)$

14:     **end for**

15: **end for**

16: **return** $\theta = \{\overrightarrow{\theta}, \overleftarrow{\theta}\}$

---

Table 4: Dataset statistics and distribution of answer set sizes.

| Dataset | Dataset Size | | Distribution of Answer Set Size | | | |
|---|---|---|---|---|---|---|
| | #Train | #Test | #Ans = 1 | $2 \leq$ #Ans $\leq 4$ | $5 \leq$ #Ans $\leq 9$ | #Ans $\geq 10$ |
| WebQSP | 2,826 | 1,628 | 51.2% | 27.4% | 8.3% | 12.1% |
| CWQ | 27,639 | 3,531 | 70.6% | 19.4% | 6.0% | 4.0% |
| GrailQA | – | 874 | 62.0% | 18.2% | 6.8% | 13.0% |

question-specific subgraphs, only answers and logical forms. Following Sun et al. (2024b), we obtain for each GrailQA question the local Freebase subgraph centered on its topic entity, and then align the subgraphs provided from Sun et al. (2024b) via the topic entity MID. For our experiments, we use a curated subset of 874 evaluation samples in the GrailQA training set, which provides 2-hop subgraphs for each topic entity.

Following previous practice, we adopt the same training and test split, with the same subgraph construction for each question-answer pair to ensure fairness (Jiang et al., 2022; Luo et al., 2024; Li et al., 2024; Mavromatis & Karypis, 2024).

# E   MORE DISCUSSION WITH RECENT KGQA FRAMEWORKS

Recent studies have treated the LLM as an agent that performs in-context search, planning, and iterative refinement over the knowledge graph (Sun et al., 2024a; Xu et al., 2024b;a; Chen et al., 2024; Li et al., 2025; Liang & Gu, 2025; Fang et al., 2025; Zhu et al., 2025; Shen et al., 2025). These approaches are largely training-free: the KG acts as an external memory, and the LLM navigates it by generating reasoning chains or plans, without training a graph retriever. For instance, Li et al. (2025) generates faithful and logically constrained reasoning chains on knowledge graphs through guided, well-formed decoding. Liang & Gu (2025) improves the breadth and depth of LLM reasoning

Table 5: Comparison with recent agent-based KGQA methods on WebQSP and CWQ.

| Method | WebQSP | | CWQ | |
|---|---|---|---|---|
| | Macro-F1 | Hit | Macro-F1 | Hit |
| DoG (Li et al., 2025) | – | 91.4 | – | 76.2 |
| GoG (Xu et al., 2024b) | – | 84.4 | – | 75.2 |
| PoG (Chen et al., 2024) | – | 87.3 | – | 75.0 |
| KARPA (Fang et al., 2025) | 72.1 | 91.2 | 61.5 | 78.4 |
| Ours (T) | 79.9 | 94.0 | 61.3 | 71.6 |

over KGs by expanding and accelerating the graph search process. Xu et al. (2024b) treats the LLM simultaneously as an agent and a KG completion module, enabling reasoning over incomplete knowledge graphs. Chen et al. (2024) introduces adaptive planning and self-correction over KG structures, iteratively refining the LLM's reasoning trajectory. Fang et al. (2025) provides a training-free mechanism to aggregate KG-derived reasoning paths as external references for LLMs. Zhu et al. (2025) incorporates self-reflective planning loops to improve the reliability and robustness of LLM-based KG reasoning. Shen et al. (2025) aligns the LLM's intermediate reasoning steps with KG evidence to strengthen consistency and correctness. Xu et al. (2024a) performs discriminative selection among KG candidates using LLM-inferred reasoning signals, without retriever training.

Our work differs from the above in that we focus on training a graph retriever rather than relying on LLM agentic planning. As can be seen in Table 5, agent-based approaches may achieve stronger performance in certain scenarios, but typically incur substantially higher computational overhead and longer inference latency. In contrast, our method provides a more computationally efficient alternative that follows a different design route, as shown in Table 6.

## F   Additional Details on Experimental Setup and Results

### F.1   Experiment Setup

For model training, we employ two 2-layer GCNs to enable bidirectional message passing. Each GCN has a hidden dimension of 512. We use the Adam optimizer (Kingma & Ba, 2014) with a learning rate of $1 \times 10^{-3}$, and a batch size of 10. Batch normalization (Ioffe & Szegedy, 2015) is not used, as we observe gradient instability when it is applied. The graph retriever is trained for 15 epochs on both datasets, and model selection is performed using cross-validation based on the validation loss. A dropout rate of 0.2 (Srivastava et al., 2014) is applied for regularization. For path-based training, when multiple valid paths exist, one is randomly selected at each training step; in triple-based learning, all triples along the ground-truth paths are treated as positives, and negatives are randomly sampled at a 1:5 positive-to-negative ratio.

**Implementation.** We utilize *networkx*(Hagberg et al., 2008) for performing line graph transformations and explore all paths between question entities (source nodes) and answer entities (target nodes), and GPT-4o is used to select rational paths for training labels. Our implementations are based on PyTorch(Paszke et al., 2019) and PyTorch Geometric (Fey & Lenssen, 2019).

### F.2   Efficiency Analysis

We compare the efficiency of `REL-RAG` and baseline methods using three metrics: average runtime, average number of LLM calls, and average number of retrieved triples. As shown in Table 6, `REL-RAG` employs a lightweight GNN-based retriever, making it inherently more efficient than agent-based RAG framework and LLM-based retriever. Compared with other GNN-based retrievers such as GNN-RAG and SubgraphRAG, `REL-RAG` achieves higher accuracy with comparable runtime and LLM calls. The improvement stems from the line-graph transformation, which introduces beneficial structural bias for capturing relation transitions.

Table 6: Efficiency analysis of different methods on WebQSP dataset.

| Methods | Hits | Avg. Runtime (s) | Avg. # LLM Calls | Avg. # Triples |
|---|---|---|---|---|
| RoG | 85.6 | 8.65 | 2 | 49 |
| ToG | 75.1 | 19.03 | 13.2 | 410 |
| GNN-RAG | 85.7 | 1.82 | 1 | 27 |
| SubgraphRAG | 90.1 | 2.63 | 1 | 100 |
| Ours | 92.4 | 1.74 | 1 | 50 |

Table 7: The impact of different label annotation methods under two training settings: path-based learning with 50 retrieved triples, and triple-based learning with 500 retrieved triples. All results are evaluated with GPT-4o-mini.

| Label Annotator | WebQSP | | CWQ | |
|---|---|---|---|---|
| | Macro-F1 | Hit | Macro-F1 | Hit |
| *Path-based Learning (50 triples)* | | | | |
| GPT-4o | **80.4** | **92.5** | **58.1** | **69.3** |
| ShortestPath | 79.8 | 92.1 | 55.7 | 66.1 |
| *Triple-based Learning (500 triples)* | | | | |
| GPT-4o | **79.9** | **94.0** | **61.3** | 71.6 |
| ShortestPath | 79.8 | 93.8 | 61.1 | **72.0** |

### F.3 ABLATION STUDY ON LABEL ANNOTATIONS

We study the effect of label annotation strategies on retriever performance. Intuitively, higher-quality annotations reduce noise and provide cleaner supervision. We compare two settings: (i) path-based learning with 50 retrieved triples, and (ii) triple-based learning with 500 retrieved triples.

As shown in Table 7, in the path-based setting, GPT-4o annotated labels yield clear gains over shortest-path supervision, showing that when the retrieval budget is limited, cleaner path labels help ensure the retrieved triples are truly relevant to the question.

In contrast, under triple-based learning with a larger retrieval budget, the performance gap nearly disappears. Although shortest-path labels are noisier, they still contain rational signals; with more retrieved triples, a strong LLM reasoner can effectively identify and exploit relevant evidence.

These results suggest that label annotation is most beneficial in low-budget retrieval scenarios, while shortest-path supervision remains sufficient when retrieval is broad and the LLM has strong reasoning capacity.

## G MOTIVATING EXAMPLES ON RATIONAL PATHS

In this section, we provide 3 intuitive examples in Figure 5 6 7 to demonstrate that not all the shortest paths are rational to the question.

## H DEMONSTRATIONS ON RETRIEVED EVIDENCE FROM REL-RAG

We provide 2 examples, with both triple-based outputs and path-based outputs, as illustrated in Figure 8 and 9 for the first example, and Figure 10 and 11 for the second example.

---

### WebQTest-923_e3a2d3d50bac69d563de83a7f72eafc0

**Question:**
Which country with religious organization leadership *Noddfa, Treorchy* borders England?

---

**Candidate shortest paths:**

```
England → location.location.adjoin_s → m.04dgsfb →
location.adjoining_relationship.adjoins → Wales                         (rational)

England → law.court_jurisdiction_area.courts → National Industrial Relations Court →
law.court.jurisdiction → Wales                                      (non-rational)

England → organization.organization_scope.organizations_with_this_scope → Police
Federation of England and Wales → organization.organization.geographic_scope → Wales
(non-rational)

England → organization.organization_scope.organizations_with_this_scope → BES Utilities
→ organization.organization.geographic_scope → Wales                (non-rational)

···
```

---

**Explanation:**
The first path directly encodes geographical adjacency, correctly identifying Wales as the country bordering England. Other paths rely on courts or organizations with overlapping scope, which do not provide evidence of territorial borders and are therefore non-rational.

Figure 5: Motivating example to illustrate that not all shortest paths are rational.

---

### WebQTest-415_b6ad66a3f1f515d0688c346e16d202e6

**Question:**
What movie with film character named Mr. Woodson did Tupac star in?

---

**Candidate shortest paths:**

```
Tupac Shakur → film.actor.film → m.0jz0c4 → film.performance.film → Gridlock'd   (rational)

Tupac Shakur → music.featured_artist.recordings → Out The Moon →
music.recording.releases → Gridlock'd                               (non-rational)

Tupac Shakur → music.featured_artist.recordings → Wanted Dead or Alive →
music.recording.releases → Gridlock'd                               (non-rational)

Tupac Shakur → music.artist.track_contributions → m.0nj8wrw →
music.track_contribution.track → Out The Moon → music.recording.releases → Gridlock'd
(non-rational)

Tupac Shakur → film.music_contributor.film → Def Jam's How to Be a Player →
film.film.produced_by → Russell Simmons → film.producer.films_executive_produced →
Gridlock'd                                                          (non-rational)
```

---

**Explanation:**
The first path models the actor–character–film linkage correctly, hence rational. Others reach the film via music or production, not by acting roles.

Figure 6: Motivating example to illustrate that not all shortest paths are rational.

## I    PROMPT TEMPLATE

We provide the prompt template in this section for rational paths filtering, as shown in Figure 12.

## J    SOFTWARE AND HARDWARE

We conduct all experiments using PyTorch (Paszke et al., 2019) (v2.1.2) and PyTorch Geometric (Fey & Lenssen, 2019) on Linux servers equipped with NVIDIA A100 GPUs (80GB) and CUDA 12.1.

---

**WebQTrn-3763_c707414f103503f2530fc654a85645fe**

**Question:**
What country close to Russia has a religious organization named *Ukrainian Greek Catholic Church*?

---

**Candidate shortest paths:**

```
Ukrainian Greek Catholic Church → religion.religious_organization.leaders → m.05tnwqd →
religion.religious_organization_leadership.jurisdiction → Ukraine                    (rational)

Russia → location.location.partially_contains → Seym River →
geography.river.basin_countries → Ukraine                                          (non-rational)

Russia → olympics.olympic_participating_country.olympics_participated_in → 2010 Winter
Olympics → olympics.olympic_games.participating_countries → Ukraine                (non-rational)

Russia → organization.organization_founder.organizations_founded → Commonwealth of
Independent States → organization.organization.founders → Ukraine                 (non-rational)

Russia → location.location.adjoin_s → m.02wj9d3 →
location.adjoining_relationship.adjoins → Ukraine                                 (non-rational)
```

---

**Explanation:**
The first path explicitly links the Ukrainian Greek Catholic Church to its jurisdiction in Ukraine, directly answering the question. The other paths connect Russia and Ukraine via geography, sports, or organizations, but do not ground the church in a jurisdiction, making them non-rational.

Figure 7: Motivating example to illustrate that not all shortest paths are rational.

---

**Case 1 (Path-based inference)**

**Question:**
Where is the *Busch Stadium* arena?

---

**Retrieved paths:**

```
Busch Stadium → location.location.containedby → St. Louis

St. Louis Cardinals → sports.sports_team.arena_stadium → Busch Stadium →
location.location.containedby → St. Louis

Busch Stadium → sports.sports_facility.home_venue_for → m.0nf2byb
→ sports.team_venue_relationship.team → St. Louis Cardinals →
sports.sports_team.arena_stadium → Busch Stadium → location.location.containedby
→ St. Louis

St. Louis → sports.sports_team_location.teams → St. Louis Cardinals →
sports.sports_team.arena_stadium → Busch Stadium

m.0nf2byb → sports.team_venue_relationship.venue → Busch Stadium →
location.location.containedby → St. Louis

2011 World Series → time.event.locations → Busch Stadium →
location.location.containedby → St. Louis

...
```

Figure 8: Path-formatted evidence for "Where is the *Busch Stadium* arena?".

## K   ETHICS STATEMENT

This work studies relation-aware retrieval for knowledge graph question answering . It does not involve human subjects, clinical data, or interventions. We use only publicly available datasets under their respective licenses and follow standard splits. No personally identifiable information is collected or generated; all evaluation uses de-identified benchmark data.

## L   LLM USAGE

We used large language models to (1) refine phrasing and improve organization of the manuscript text, (2) draft and refactor parts of the experimental code, and (3) explore and sanity-check concepts in learning theory.

---

**Case 1 (Triple-based inference)**

**Question:**
Where is the Busch Stadium arena?

---

**Retrieved triples:**

```
Busch Stadium → location.location.containedby → St. Louis

St. Louis Cardinals → sports.sports_team.arena_stadium → Busch Stadium

Busch Stadium → sports.sports_facility.home_venue_for → m.0nf2byb

m.0nf2byb → sports.team_venue_relationship.team → St. Louis Cardinals

m.0nf2byb → sports.team_venue_relationship.venue → Busch Stadium

St. Louis → sports.sports_team_location.teams → St. Louis Cardinals

2011 World Series → time.event.locations → Busch Stadium

2006 World Series → sports.sports_championship_event.champion → St. Louis Cardinals

Roger Dean Stadium → sports.sports_facility.teams → St. Louis Cardinals

Busch Stadium → location.location.events → 2011 World Series

...
```

Figure 9: Triple-formatted evidence for "Where is the Busch Stadium arena?"

---

**Case 2 (Path-based inference)**

**Question:**
In which movies does Logan Lerman act that were production designed by Andrew Menzies?

---

**Retrieved paths:**

```
Logan Lerman → film.actor.film → m.03jq2p4 → film.performance.film → 3:10 to Yuma

Andrew Menzies → film.film_production_designer.films_production_designed → 3:10 to Yuma

Logan Lerman → film.actor.film → m.0v4n3ld → film.performance.film → Fury

Andrew Menzies → film.film_production_designer.films_production_designed → Fury

Logan Lerman → film.actor.film → m.0cfzvvt → film.performance.film → The Perks of
Being a Wallflower

Andrew Menzies → film.film_production_designer.films_production_designed → Knight and
Day

Andrew Menzies → film.film_production_designer.films_production_designed → Identity

Logan Lerman → film.actor.film → m.0646ffc → film.performance.film → Percy Jackson &
the Olympians:  The Lightning Thief

...
```

Figure 10: Path-formatted evidence for "In which movies does Logan Lerman act that were production designed by Andrew Menzies?".

---

**Case 2 (Triple-based inference)**

**Question:**
In which movies does Logan Lerman act that were production designed by Andrew Menzies?

---

**Retrieved triples:**

Andrew Menzies → film.film_production_designer.films_production_designed → Fury

Logan Lerman → film.actor.film → m.0v4n3ld
m.0v4n3ld → film.performance.film → Fury

Andrew Menzies → film.film_production_designer.films_production_designed → 3:10 to Yuma

Logan Lerman → film.actor.film → m.03jq2p4
m.03jq2p4 → film.performance.film → 3:10 to Yuma

Andrew Menzies → film.film_production_designer.films_production_designed → Knight and Day

Andrew Menzies → film.film_production_designer.films_production_designed → Identity

Andrew Menzies → film.film_production_designer.films_production_designed → The Uninvited

Andrew Menzies → film.film_production_designer.films_production_designed → G.I. Joe: Retaliation

Logan Lerman → film.actor.film → m.0cfzvvt
m.0cfzvvt → film.performance.film → The Perks of Being a Wallflower

...

Figure 11: Triple-formatted evidence for "In which movies does Logan Lerman act that were production designed by Andrew Menzies?"

---

**Prompt template for identifying rational paths**

**Example**

Given a question <example question>, the reasoning paths are:

<reasoning paths>

The rational paths are:

<Rational Paths>

**Explanation**

<Explanation>

**Task**

Now given question <question>, the reasoning paths are:

<Candidate Paths>

Identify all the rational paths, and list below, with explanations:

<Rational Paths>

<Explanations>

---

Figure 12: Prompt template for retrieving rational reasoning paths.

