# OpenReview forum: "REL-RAG: Relation-Aware Retrieval-Augmented Generation for Generalizable Knowledge Graph Question Answering"
_ICLR.cc/2026/Conference — Submitted to ICLR 2026_

### Official Review · Reviewer_Lrvk · 2025-10-22

**Soundness:** 3
**Presentation:** 1
**Contribution:** 2
**Rating:** 2
**Confidence:** 4

**Summary:**

This paper proposes REL-RAG, a GNN-based RAG solution for knowledge graph question answering.
The main contribution of REL-RAG lies in follows. First, REL-RAG proposes to transform KG into a line graph, which each triple is a node. The REL-RAG then devises a message passing mechanism on this line graph. From the theoretical perspective, the authors prove that owing to line-graph, GNN retriever has better generalization. The experimental results also empirically show the effectiveness of the proposed method.

**Strengths:**

1.	The proposed method shows to be effective, achieving better performance than baseline method Sub-graph RAG and GNN-RAG.

2.	The authors theoretically proves the better generalization of the proposed method, making the main claims theoretically grounded.

3.	The manuscript shows to have comprehensive experimental design, which empirically reconfirms the conclusion by testing OOD performance.

**Weaknesses:**

1.	This paper contains several confusions need to clearly defined or explained.

2.	The paper states how to perform message passing and path encoding in section 3, but does not introduce how to select out the best relation path that contributes to the question. To be specific, there exists a clear logic gap between Section 3 and Section 4. This reviewer cannot even make an educatable guess that how z_t^{(1)}  in equation (5) contributes in equation (10) and (11).

3.	In Appendix E.1, the manuscript mentions “when multiple valid paths exist”. However, in Section 4 and Appendix B (Inference.), how to select valid path in path-based learning is not explicitly mentioned. This reviewer can only make an educatable guess that we can calculate the product of each triple within the path and sort them. In addition, the term “predicted probabilities” mentioned in line 1216/1217, lack clear definition. This reviewer can only make an educatable guess that the probability of a triple is the softmax term exp(z_q, z_q(i))/ \sum … in equation (75).

4.	Based on the available information provided in this manuscript, it can be hard for an educatable researcher to construct the system and replicate the results. Due to the logical gaps and the absence of precise definitions, this reviewer believes that researchers outside the KBQA-related areas may be completely lost when reading this paper.

5.	This paper lacks discussion and/or comparison with several baseline methods, namely DoG [1], FastToG [2], GoG [3], PoG [4], KARPA [5], SRP [6], RAR [7], and READS [8], where [1-5] are accepted to top-tier conferences.

[1] Li et al., Decoding on Graphs: Faithful and Sound Reasoning on Knowledge Graphs through Generation of Well-Formed Chains (ACL2025)

[2] Liang and Gu., Fast Think-on-Graph: Wider, Deeper and Faster Reasoning of Large Language Model on Knowledge Graph (AAAI2025)

[3] Xu et al., Generate-on-Graph: Treat LLM as both Agent and KG for Incomplete Knowledge Graph Question Answering (EMNLP2024)

[4] Chen et al., Plan-on-Graph: Self-Correcting Adaptive Planning of Large Language Model on Knowledge Graphs (NeurIPS 2024)

[5] Fang et al., KARPA: A Training-free Method of Adapting Knowledge Graph as References for Large Language Model’s Reasoning Path Aggregation (ACL 2025)

[6] Zhu et al., Self-Reflective Planning with Knowledge Graphs: Enhancing LLM Reasoning Reliability for Question Answering

[7] Shen et al., Reason-Align-Respond: Aligning LLM Reasoning with Knowledge Graphs for KGQA

[8] Xu et al., LLM-based Discriminative Reasoning for Knowledge Graph Question Answering

**Questions:**

1.	What does “First-class objects” means needs to be clearly defined in the manuscript. It appears 4 times in the manuscript but neither of its occurrences is associated with clear definitions / citation to other paper.

2.	Although theorems and proofs can lend rigor and credibility to the main claims, ***their inclusion should not come at the expense of the main sections’ overall logical coherence***. According to W2, this reviewer strongly suggests the authors to move equation (74)-(76) to the main sections and provide more detailed elaboration.

3.	Which specific work does “prior work” in line 354 means? Are embeddings of entities and relations learnable? For a “representation learning” conference, these are highly important. This reviewer sincerely requests the authors to add relevant information, including explanations and citations.

4.	This reviewer sincerely requests the authors to discuss related works [1]-[8] mentioned before, and compare [1]-[5] accordingly.

In view of the manuscript’s current organization, this reviewer recommends rejection. ***Nonetheless, given the paper’s findings and contributions, this reviewer would consider raising the score contingent on a comprehensive improvement of the overall organization***.

---

> ### Author Response · Authors · 2025-11-21
> **Reply to Reviewer Lrvk**
>
> Thank you for your careful review and insightful comments. Please see below for our responses to your comments and concerns.
>
> ---
>
> > **W1,W2 & Q2: Intention of Section 3 and logical coherence**
>
> We thank the reviewer for highlighting the need to clarify the connection between Section 3 and Section 4.
>
> **The role of Equation (5).**  Equation (5) plays a theoretical role. It shows that line graph MPNN incurs a significantly simpler solution for learning multi-hop relational paths than message passing on the raw graph, where all relation embeddings are intertwined at entity nodes. This result provides the theoretical motivation for why line graphs are beneficial in KGQA.
>
> **Connection to implementation.**  We wish to clarify that Equation (5) is not directly used in the derivation of Equations (10) and (11). Instead, the simplified solution space motivates the overall line graph–based retriever design in Section 4, which makes the practical learning problem easier and **implicitly** strengthens the performance of Equations (10) and (11).
>
> In summary, we clarify that Eq. 5 illustrates the data-centric benefit introduced by the line-graph representation for learning multi-hop relation paths. In the implementation, we do not modify the model architecture nor introduce any additional regularizers.
>
> **Revision.** We have revised Section 3 and Section 4 (highlighted in blue) to more clearly distinguish the theoretical motivation from the practical implementation, and make the technical approach more detailed to improve the logical coherence.
>
> ---
>
> > **W3: Path Selection and Predicted Probabilities in Path-based Learning**
>
> **Path selection for path-based learning.**  For path-based learning, we first compute the shortest paths between the question entity and the answer entity. These candidate paths are passed to an LLM, which determines which paths are semantically relevant to the question; these selected paths then serve as supervision labels.
>
> **Definition of predicted probabilities:**  The predicted probabilities are obtained by ranking triples according to their scores $⟨z_q, z_v⟩$, from which the top-k triples are selected.
>
> We have updated Appendix B and Section 4 in our revision to add more details and improve clarity.
>
> > **W4: Reproducibility and Clarity for Broader Audience**
>
> To address the reviewer's concern, we have revised Section 3 and Section 4 to include more detailed technical explanations of our method, ensuring that the approach can be readily reproduced by broader research communities. We will also release all of the code and data to further support reproducibility.
>
>
> > **W5 & Q5: Discussion and Comparison with Additional Baseline Methods**
>
> We appreciate the reviewer pointing us to these recent works. We have added them to the revised manuscript and included an detailed discussion in Appendix E.
>
> These works [1]–[8] generally treat the LLM as an agent that performs in-context KG navigation through search, planning, and iterative refinement. They are typically training-free: the KG serves as an external memory, and the LLM generates reasoning chains or plans without training a dedicated graph retriever.
>
> Our work differs in that we train a graph retriever rather than relying on agentic planning. We acknowledge that agent-based approaches may achieve stronger performance but often incur higher computational cost and longer inference time. Our method offers a more computationally efficient alternative that takes a different direction.
>
> > **Q1 & Q2: The Terminology “First-class Object” and Overall Coherence**
>
> We thank the reviewer for pointing out the need for clearer terminology and improved coherence.
>
> We now explicitly define “first-class objects” at its first occurrence (lines 68–69) to avoid ambiguity.
>
> Additionally, we have revised Section 3 and 4. These changes address previous logical gaps and ensure smoother progression from theoretical motivation to practical implementation, and provide more technical details to improve clarity.
>
> ---
>
> > **Q3: Prior Work Citation and Embedding Learning Details**
>
> We have added the appropriate citations to prior work referenced. Moreover, we now clearly state that entity and relation embeddings are frozen during training (line 416), clarifying this implementation detail.
>
> ---
>
> We sincerely thank the reviewer for the careful review and constructive feedback. We hope that our responses have addressed all concerns.

---

> > ### Comment · Reviewer_Lrvk · 2025-11-24
> > **Reply to the Rebuttal**
> >
> > This reviewer appreciates the efforts made by the authors in providing relevant explanations and revising the manuscript accordingly.
> >
> > However, this reviewer still has some doubts that need to be addressed. The reviewer also believes such doubts will confuse potential readers.
> >
> > Regarding W1, the authors state that *"***Equation (5) is not directly used***  in the derivation of Equations (10) and (11)"***.
> > This statement ***contradicts*** the other statement in the rebuttal, namely *"we clarify that ***Eq. 5*** illustrates the data-centric benefit introduced by the line-graph representation for learning multi-hop relation paths. In the implementation, we ***do not modify the model architecture*** nor introduce any additional regularizers.*
> >
> > Therefore, this reviewer is still uncertain whether $f_\theta$ in Line 202-203 follows the ***same*** generic message passing process introduced in Eq. (3-5). Based on the revised manuscript and the contradictory statements above, one is not able to figure out how the 2-layer MPNN model is implemented, and how we can implement $f_\theta$ and calculate $z_q$, $z_{q(i)}$ and $z_j$.
> >
> > Regarding reply W3: The second issue mentioned in my original W3 is not addressed. Specifically, what does the term "probabilities" in line 337 of the revised manuscript means? This reviewer did not find any clue referring to the "probability" from line 278 to line 319, and hence, sincerely requests the authors provide the formal definition.

---

> ### Author Response · Authors · 2025-11-25
> **Reply to Reviewer Lrvk**
>
> We sincerely thank the reviewer for the continued careful review and for raising these important clarification points. We apologize for the confusion caused by our imprecise wording in the previous rebuttal.
>
> >  ### **Implementation of f_θ and MPNN**
>
> We acknowledge that our previous statement was misleading. To clarify: **Equations 3, 4, and 5 are indeed used in our implementation**. The process is as follows:
>
> 1. For each subgraph $\mathcal{G}\_q$, we first apply the line graph transformation to obtain $\mathcal{G}'_q$.
> 2. We then run the generic MPNN update (Equations 3-5) on $\mathcal{G}'_q$ to obtain the node representations $\mathbf{z}\_{q(i)}$ and $\mathbf{z}\_j$, where $j \in \\mathcal{N}(q(i))$ denotes the neighbors of node $q(i)$ in the line graph, with an additional *stop* node. For $\mathbf{z}\_{q}$, it is obtained from the text encoder followed by a linear layer to keep aligned with the hidden dimensions with $\mathbf{z}\_{q(i)}$ and $\mathbf{z}\_j$.
> 3. $f_\theta$  is implemented as a standard MPNN (we use GCN in our experiments) that directly follows the message passing framework defined in Equations 3-5.
>
>
> > ### **Definition of "probabilities"**
>
> We apologize for the lack of clarity. The term "probabilities" in line 337 refers to the softmax probabilities computed during inference. Specifically:
>
> During inference with path-based learning, after obtaining the initial triple node $v\_{q(0)}$ via Equation 12, we compute the probability of each neighboring node using the softmax function derived from Equation 11:
>
> $$
> P(v\_j | v\_{q(i-1)}, q) = \\frac{\\exp(\langle \mathbf{z}\_q, \mathbf{z}\_j \rangle)}{\\sum\_{k \in \mathcal{N}(q(i-1))} \\exp(\langle \mathbf{z}\_q, \mathbf{z}\_k \rangle)}
> $$
>
> We then sample the next hop according to this probability distribution, repeating until a stop node is sampled or the maximum depth is reached. We have updated the draft in **lines 302-304 and line 1257-1259** to explicitly describe this inference procedure, and to improve clarity.

---

### Official Review · Reviewer_Sq8a · 2025-10-25

**Soundness:** 2
**Presentation:** 2
**Contribution:** 2
**Rating:** 4
**Confidence:** 4

**Summary:**

REL-RAG proposes a line graph transformation that better adapts knowledge graphs for message passing in KGQA tasks. By treating triplets as nodes in this transformation, REL-RAG enables two learning strategies: sequential path-based learning and classic node classification (equivalent to triplet retrieval). Experimental results on WebQSP and CWQ benchmarks demonstrate that REL-RAG achieves competitive performance against other graph retrievers, with the line graph showing better generalization compared to the original graph structure.

**Strengths:**

- S1) REL-RAG operates on a line graph that treats triplets as nodes, better suiting triplet retrieval with GNNs. By compacting triplets into graph nodes, the underlying GNN requires fewer layers to work effectively and the line graph transformation enables REL-RAG to employ GNNs for both path-based learning (sequential predictions) and triplet retrieval tasks (Section 4).

- S2) Table 2 demonstrates that the line graph achieves better generalization compared to the raw graph. This improvement likely stems from triplets having richer semantics than individual nodes, making unseen triplets easier to handle than unseen nodes.

**Weaknesses:**

- W1) REL-RAG's theoretical analysis (Section 3) assumes GCN-style MPNNs that aggregate information from all relations (Eq. 1). However, practical GNN/MPNN implementations use query-conditioned message passing based on semantic similarity between the query and relations/subgraphs (NSM, GNN-RAG), which the current theory doesn't capture. This limits the insights to a narrow case; the authors should also demonstrate that line graph transformation benefits other GNN architectures beyond GCN.

- W2) REL-RAG requires storing embeddings for all triplets (edges), making it computationally impractical for billion-scale or real-world graphs. While REL-RAG's performance may benefit from triplets
generalizing better than graph nodes (e.g., handling unseen entity names), its practical application is limited its storage requirements.


-  W3) The paper makes some overclaims on performance results. Line 369 states "REL-RAG achieves improvements of up to 20.3% on CWQ," but the actual results show GNN-RAG at 66.8% versus REL-RAG at 67.2%, which is only a 0.4 percentage point difference. Given that REL-RAG retrieves more triplets than GNN-RAG (Table 5), these improvements appear marginal.

**Questions:**

- Q1) How is the question triplet $v_{q(0)}$ obtained in Section 4? This appears to be a crucial component but I could not find the explanation.

- Q2) Can the line graph transformation benefit other KGQA GNNs beyond vanilla GCN?

---

> ### Author Response · Authors · 2025-11-21
> **Reply to Reviewer Sq8a**
>
> We thank the reviewer for the constructive feedback and address the concern below.
>
> ---
>
> > **W1: Theoretical Analysis Under Query-Conditioned Message Passing**
>
> We acknowledge the reviewer’s point regarding query-conditioned message passing and clarify that our theoretical argument remains valid even when incorporating query-dependent relation weights $\omega(q,r)$.
>
> **Theoretical justification.** The raw-graph MPNN update with query-conditioned weighting is:
>
> $$
> h_v^{(l+1)} = \\phi^{(l)}\\!\\left(h_v^{(l)}, \\sum_{u\\in\\mathcal{N}(v)} \\omega(q,r_{vu})\\, m_{v\\leftarrow u}^{(l)}\\right).
> $$
>
> Consider a 2-hop reasoning chain $r_1 \\rightarrow r\_2$ relevant to query $q$. If the first hop is optimally aligned (i.e., $\\omega(q,r\_1) = 1$ and $\\omega(q,r\_{\\text{other}})=0$), then the query embedding $z_q$ becomes aligned with the embedding of $r\_1$.
>
> Consequently, at the second hop we generally have $\\omega(q,r\_2) \\neq 1$, since $z\_q$ is no longer equally aligned with $r_2$. As a result, the intermediate entity node still aggregates messages from multiple incident relations with non-zero weights, making it difficult for the model to learn the intended composition $r\_1 \\!\\to\\! r\_2$.
>
> Therefore, we respectively argue that **our theoretical insight remains valid**: relation mixing on raw graphs persists even under query-conditioned weighting.
>
>
> > **W2: Computational Practicality and Storage Requirements**
>
> We appreciate the reviewer raising this important concern. We wish to clarify that REL-RAG does not operate on the **full knowledge graph**, which would indeed be impractical for billion-scale graphs. Instead, following standard pipeline, we first extract a query-specific subgraph for each question by linking question entities and retrieving their k-hop neighborhoods. The line graph transformation is then applied only to this subgraph, which typically contains hundreds to thousands of triples rather than billions.
>
>
> This design ensures computational feasibility. To address the reviewer's concern about storage and runtime, we provide empirical measurements of memory usage and wall-clock time on the two datasets:
>
> **Table: Runtime and memory cost at inference time with GCN and GAT as graph retriever**
> | Model | WebQSP (avg memory / avg runtime) | CWQ (avg memory / avg runtime) |
> |-------|-----------------------------------|--------------------------------|
> | GCN   | 898MB / 2.71s                  | 1207MB / 3.18s               |
> | GAT   | 2512MB / 2.18s                  | 3511MB / 3.66s               |
>
> As shown, the memory and computational overhead remains manageable for practical deployment. The **subgraph-based** approach makes REL-RAG applicable to real-world knowledge graphs of any scale.
>
> > **W3: Clarification of performance gain**
>
> We thank the reviewer for the careful observation. The "up to 20.3%" improvement refers to the gain over SubgraphRAG + Llama3.1-8B. GNN-RAG uses a **finetuned** Llama model on CWQ’s 20,000+ training examples, making direct comparison inappropriate. We have clarified this distinction in Section 5.2 of the revised manuscript.
>
> > **Q1: How to obtain initial triple $v_{q(0)}$**
>
> We thank the reviewer for raising this important point. We have updated Section 4 of the manuscript to clarify how $v\_{q(0)}$ is selected.
> Specifically, a negative sampling loss (Eq. 14) is utilized to optimize the graph retriever to select $v_{q(0)}$, and during inference stage, we sample the initial question entity using the score $\\langle z_q,z_v \\rangle$. More details of the approach can be seen in Section 4.1 in our revision.
>
> > **Q2: Can the line graph transformation benefit other KGQA GNNs beyond vanilla GCN?$**
>
> As shown below, applying the line-graph transformation to GAT and SGConv yields performance comparable to or even slightly better than GCN. These results indicate that the benefit of the line-graph representation is **not limited to a specific GNN architecture**.
>
> **Table: REL-RAG with different types of GNN architectures**
> | Method                      | CWQ Macro-F1 | CWQ Hit | GrailQA Macro-F1 | GrailQA Hit |
> |-----------------------------|-------------:|--------:|-----------------:|------------:|
> | Raw graph                   | 51.78        | 62.84   | 32.99            | 49.67       |
> | Line graph (GCN, ours)      | 57.00        | 67.52   | 34.68            | 52.42       |
> | Line graph (GAT)            | 58.55        | 68.85   | 34.64            | 52.38       |
> | Line graph (SGConv)         | 57.85        | 68.39   | 34.23            | 51.34       |
>
> ---
>
> We sincerely thank you for your careful review. We hope we have addressed your concerns of our approach.

---

> > ### Comment · Reviewer_Sq8a · 2025-11-26
> >
> > Thank you for the discussion and the effort to address my questions, such extending the study to more GNNs. I still have some concerns regarding the applicability of the approach (theory & computational cost) in realistic scenarios.
> >
> > > W1: Theoretical Analysis Under Query-Conditioned Message Passing
> >
> > Following your response, $\omega(q,r)$ can be an embedding similarity function able to yield high matching probabilities with more than one relations. Thus, the claim "since $z_q$ is no longer equally aligned with $r_2$ at the second hop" does not seem realistic.
> >
> > This also relates to Q2, and whether line graph transformation is helpful for GNN that already implement a version of $\omega(q,r)$, e.g., NSM (He et al., 2021).
> >
> > > W2 & Q1
> >
> > Computing triplet embeddings can be expensive for dense subgraphs, as in a scenario where a `product` node is connected to many `customer` nodes, e.g. `(productA, bought_by, {customer1, ..., customer1,000})`. I am wondering whether the method would scale in those scenarios, such as linking $v_{q^0}$ via triplet embedding.
> >
> > > W3: Clarification of performance gain
> >
> > REL-RAG uses more modern LLMs (Llama-3.1-8B, GPT-4o-mini) than GNN-RAG, thus a more appropriate comparison with same LLMs would be suggested.

---

> ### Author Response · Authors · 2025-11-28
>
> > ### **W1: Theoretical Analysis Under Query-Conditioned Message Passing**
>
> We thank the reviewer for this important question. We clarify why line graphs remain beneficial even with query-conditioned weighting functions like $\\omega(q, r)$.
>
> **The Core Issue: Relation Mixing.** The fundamental advantage of line graphs is structural elimination of relation embedding mixing. In entity graphs, MPNN updates aggregate messages from multiple neighbors with different relations simultaneously, mixing signals from competing paths. Line graphs architecturally separate relation transitions into distinct nodes, eliminating this mixing.
>
> **Why Query-Conditioning Cannot Replicate This.** For $\omega(q, r)$ to achieve equivalent separation, it would need perfect gating: $\omega(q, r\_1)=1$, $\omega(q, r\_2)=1$ for ground-truth path $r\_1 \rightarrow r\_2$, and $\omega(q, r)=0$ for all other relations. This is geometrically impossible with similarity-based matching functions.
>
> Consider NSM's implementation: $s(q, r) = f(q)^{\top} h(r)$ where $f$ and $h$ are RoBERTa encoders. If we align $q$ with $r\_1$ to maximize $\omega(q, r\_1)$, then $q$ necessarily misaligns with $r\_2$, yielding $\omega(q, r\_2)<1$. Furthermore, semantically similar relations receive non-zero weights due to **continuous embedding spaces**. The soft nature of learned similarity functions cannot produce the hard binary gating needed to eliminate relation mixing.
>
> Therefore, even with query-conditioned weighting $\\omega(q, r)$ in MPNN updates, **the relation mixing problem persists**, which is precisely the core advantage that line graph transformation addresses.
>
>
> > ### **W2 & Q1: Scalability of Line Graph Transformation in Dense Subgraphs**
>
> We thank the reviewer for raising this important scalability concern. To address this concern, we conducted experiments and analysis:
>
> **Scalability Experiments.** We use networkx to generate 100 synthetic graphs, each containing 2,000 nodes with varying average degrees (10, 20, and 30) to simulate different density scenarios. To test the method under skewed distributions, we deliberately introduced high-degree nodes with maximum degree of 1000 (affecting approximately 3% of nodes). We performed line graph transformation and measured both graph generation time and memory consumption. The results are summarized below:
>
> | Avg Degree | Line Graph Generation Time | Memory Usage |
> |------------|---------------------------|--------------|
> | 5         | 0.30 sec                  | 22.77 MB     |
> | 10         | 1.07 sec                  | 80.00 MB     |
> | 20         | 4.01 sec                  | 297.62 MB    |
> | 30         | 7.74 sec                  | 651.70 MB    |
>
> These results demonstrate that the computational cost of line graph transformation depends primarily on the **average degree** rather than the presence of skewed high-degree nodes. Memory consumption remains manageable across all tested configurations, staying well below 1 GB even for graphs with average degree of 30. The processing time becomes more substantial at higher average degrees (approximately 7.74 seconds for average degree 30).
>
> **Real-World Dataset Characteristics.** We further analyzed the subgraph statistics from the WebQSP and CWQ benchmarks used in our experiments:
>
> | Dataset | Avg Nodes | Avg Edges | Avg Degree | Max Degree |
> |---------|-----------|-----------|------------|------------|
> | WebQSP  | 1388.82  | 3919.46  | 5.64       | 2841      |
> | CWQ     | 1281.04  | 3852.49  | 6.01       | 2773      |
>
> While both datasets contain high-degree hub nodes (maximum degrees exceeding 2,700), the overall average degree remains low (5.64 for WebQSP and 6.01 for CWQ). Therefore, line graph transformation remains computationally practical for subgraph-based KBQA scenarios even with some hub nodes with high degrees.
>
> > ### **W3: Performance comparison**
>
> We agree with the reviewer that comparing REL-RAG and GNN-RAG with the same LLMs would provide a fairer assessment of our contribution. Due to limited time during the rebuttal period, we will add results with the same LLM configurations in the future revision.

---

> > ### Author Response · Authors · 2025-12-01
> > **Performance comparison with GNN-RAG**
> >
> > > ### **Response to W3: Clarification of performance gain under matched LLM settings**
> >
> > We have used the official GNN-RAG codebase and re-ran their retriever with the same inference setup as REL-RAG including:
> >
> > - **LLM**: Llama-3.1-8B-Instruct
> > - **Token budget**: 50 triples
> > - Same evaluation pipeline
> >
> > The results are as follows:
> >
> > | Dataset         | Method   | F1     | Hit    |
> > |-----------------|----------|--------|--------|
> > | **CWQ**         | GNN-RAG  | 0.5242 | 0.6391 |
> > |                 | REL-RAG  | 0.5680 | 0.6720 |
> > | **WebQSP → CWQ**| GNN-RAG  | 0.4438 | 0.5597 |
> > |                 | REL-RAG  | 0.5360 | 0.6410 |
> >
> >
> > These results show substantial improvements from our framework, with up to **+9.3%** $\\uparrow$ in cross-dataset setting, and up to **+4.4%** $\\uparrow$ in in-distribution setting.
> >
> > The results imply that NSM and weighted message-passing does not address the **relation-mixing issue**, which can be addressed by the structural bias from line graph transformation.

---

### Official Review · Reviewer_o1im · 2025-10-28

**Soundness:** 3
**Presentation:** 3
**Contribution:** 3
**Rating:** 6
**Confidence:** 4

**Summary:**

This paper introduces REL-RAG, a relation-aware retrieval-augmented generation framework for knowledge graph question answering (KGQA). The key innovation is to transfer KG into line graph, elevating relations to first-class objects and explicitly encoding relation transitions. This paper provides a thorough theoretical analysis showing that this transformation leads to tighter generalization bounds compared to standard entity-graph representations. The model offers two training regimes: path-based learning for token-efficient reasoning with smaller LLM and triple-based learning for evidence-diverse reasoning with larger LLMs, which makes the contribution even larger and model more flexible. Experiments on WebQSP, CWQ, and GraiLQA benchmarks demonstrate improvements over existing baselines.

**Strengths:**

1. The paper is well written, with clear motivation and problem formulation, and provides a compelling intuition for why line graph transformation helps - by eliminating "relational mixing" at entity nodes and making relation transitions explicit in the graph structure. The visual illustrations effectively communicate this idea. The paper goes beyond empirical results to provide formal analysis, including proofs that line graph models admit tighter generalization bounds under various distribution shifts (ID, compositional, and OOD). This theoretical foundation strengthens the contribution.

2. Transfer KG into line-graph sounds interesting and has been proven to have good results.

3. The dual training regime (path-based vs. triple-based) shows consideration for real-world deployment with different LLM capacities and token budgets.

4. The experiments cover multiple datasets and different types of generalization scenarios.

**Weaknesses:**

1. As for theoretical analysis, the proofs rely on several strong assumptions (e.g., sub-Gaussian concentration, specific Lipschitz constants) that may not hold in practice. The connection between the theoretical guarantees and empirical performance is not clearly established.

2. While the method achieves SOTA results, the improvements are often modest (e.g., 3.1% on WebQSP with 500 triples). Given the additional complexity of line graph transformation, the cost-benefit trade-off is questionable.

3. The paper mentions $O(|E|d_{max})$ preprocessing time but doesn't provide wall-clock comparisons or memory usage analysis. For large-scale KGs, this could be a significant limitation.

4. Some design decisions lack justification (e.g., why specifically 50 vs 500 triples? Why 2-layer GCN?). The reliance on GPT-4o for path annotation introduces a circular dependency that isn't adequately addressed.

**Questions:**

1. Have you evaluated the scalability of line graph transformation on industrial-scale KGs (e.g., full Freebase or Wikidata)? What are the practical memory and time constraints?
2. Can you provide examples where line graph transformation actually hurts performance? Are there specific types of questions or graph structures where the approach fails?
3. The theoretical analysis assumes the number of relations R ≤ d (embedding dimension). How realistic is this assumption for real-world KGs, and what happens when it's violated? (In ultra-large KG, the number of relations may be larger than embedding dimension)
4. Why 50 triples are used? And why 2-layer GCN used? Can you provide some sensitivity analysis to justify your choice?
5. The improvement on GrailQA is relatively small compared to other datasets. However, GrailQA has the smallest value, suggesting significant room for improvement. Can you provide some analysis to explain why this is the case?
6. How sensitive is the method to the quality of the initial shortest path extraction? Have you experimented with alternative path selection strategies?

---

> ### Author Response · Authors · 2025-11-21
> **Reply to Reviewer o1im**
>
> Thank you for the insightful comments and positive feedback. Please find our responses below.
>
> ---
>
> > **W1: Theoretical Assumptions and Their Practical Relevance**
>
> We thank the reviewer for raising this concern. We wish to clarify that the assumptions used in our theoretical analysis, such as sub-Gaussian concentration and bounded Lipschitz constants, are standard and well-aligned with practical GNN behavior on knowledge graphs.
>
> **Theoretical assumptions.** When aggregating over many training instances, the Central Limit Theorem implies that learned parameters tend toward approximately Gaussian distributions, and sub-Gaussian tails serve as a natural relaxation. Similarly, Lipschitz continuity is inherent to GNN architectures: spectral normalization in GCN layers and 1-Lipschitz activation functions (e.g., ReLU, sigmoid) ensure bounded Lipschitz constants by construction. These properties are not merely convenient assumptions but fundamental characteristics of modern GNNs.
>
> **Link between theory and practice.** Regarding the link between theory and empirical performance, our predictions are supported by the **zero-shot transfer experiments**. As shown in Table 2 in our draft, adopting the line-graph representation consistently improves generalization across compositional and OOD shifts while keeping all other factors fixed. For example, the WebQSP to CWQ transfer (primarily compositional shift) shows gains of 10.1% Macro-F1 and 7.5% Hits. The WebQSP to GrailQA transfer presents both compositional and OOD shifts, as GrailQA covers substantially different domains with more diverse question types and relation patterns not seen in WebQSP's primarily single-hop questions. Despite this more challenging scenario, the line graph still achieves 5.12% and 5.54% improvements over raw graph in Macro-F1 and Hits respectively.
>
> > **W2 & W3 & Q1: Cost–Benefit Trade-off of Line Graph Transformation and Preprocessing Time**
>
> We appreciate the reviewer’s concern regarding the practical trade-off between performance gains and computational overhead. Importantly, the line-graph transformation in REL-RAG is applied **only to the extracted subgraph for each query**, rather than to the entire KG.
>
> To further address this concern, we report the processing time and memory measurements on WebQSP. As shown below, the transformation incurs modest overhead and memory usage per query:
>
> **Table: Time and memory cost for line graph preprocessing.**
> | Metric                               | Value        |
> |--------------------------------------|--------------|
> | Average time per graph (s)           | 0.586        |
> | Max time per graph (s)               | 3.530        |
> | Average cuda memory per graph (MB)  | 103.81       |
> | Max cuda memory per graph (MB)      | 623.41       |
>
> Together with the observed improvements, the empirical benefits outweigh the modest preprocessing cost. We will include these results in the revised manuscript.
>
> > **W4 & Q4: Design Choices and use of GPT-4o for Path Annotation**
>
> Regarding the choice of 50 vs. 500 triples, our goal is to ensure a **fair and controlled comparison** with Subgraph-RAG, which is the closest baseline that also designs a graph retriever specifically for KGQA and uses the same LLM for answering questions. Subgraph-RAG employs a 100-triple setting; in contrast, we adopt a 50-triple configuration to demonstrate that under our proposed path-based learning regime, the retriever can achieve clearly superior performance while using only **half of the token budget**. The 500-triple setting is included to match Subgraph-RAG’s setting for a fair comparison.
>
> For the question on *why specifically a 2-layer GCN*, we experimented with deeper architectures. The 3-layer GCN yielded marginal or no improvement and occasionally underperformed. A likely reason is that, in the line-graph view, **a 2-layer GCN already covers up to 3-hop relational patterns**, which matches the dominant reasoning depth in these datasets; thus deeper GCNs do not provide additional benefit.
>
> Below we report results on CWQ (500 triples with GPT-4o-mini):
>
> **Table: Performance under different GCN depths**
> | GCN Depth | Macro-F1 | Hits@1 |
> |-----------|-----------|--------|
> | 2-layer   | 56.1      | 67.6   |
> | 3-layer   | 56.0      | 67.8   |
>
> Concerning the comment on GPT-4o and potential circular dependency: GPT-4o is used **only to select a subset of high-quality training paths**, not to answer questions or generate supervision signals that could leak into evaluation. Moreover, we evaluate REL-RAG in **zero-shot transfer settings** across CWQ and GrailQA, where GPT-4o-based path selection from WebQSP cannot provide any dataset-specific advantage. These results confirm that the improvement stems from the methodology rather than any dependency on GPT-4o.

---

> ### Author Response · Authors · 2025-11-21
> **Reply to Reviewer o1im - Part 2**
>
> > **Q2: Cases Where Line-Graph Transformation May Hurt Performance**
>
> We thank the reviewer for this thoughtful question. We identify scenarios where the combination of line graphs with deeper GNN architectures can lead to performance drops in WebQSP dataset. For example, on WebQSP, using a 3-layer GCN results in slightly **worse** performance compared to a 2-layer GCN. A plausible explanation is that most WebQSP questions are 1-hop; with the line-graph representation, a 3-layer GCN effectively expands the receptive field to 4-hop transitions, which may cause the model to learn spurious relation transitions. This may be a limitation of the *retriever architecture*, rather than a limitation arising from the line-graph transformation itself.
>
> > **Q3: Response to the $R \le d$ Assumption**
>
> Thank you for this important question. In our experimental setting with WebQSP and CWQ datasets, the assumption $R \le d$ is reasonable. These datasets use curated subsets of Freebase relations relevant to their question domains, and our hidden dimension of 512 can accommodates the number of unique relations typically encountered in these benchmarks.
>
> Even when this assumption is violated in larger-scale knowledge graphs, our core theoretical conclusions remain valid, only the specific form of the bounds changes slightly. When $R \le d$, the parameter space for the line graph model becomes $\mathcal W_{G'} \subset \mathbb R^{d^2 + R d}$, and we can no longer simplify to $O(d^2)$. The covering number becomes:
>
> $$
> \log \mathcal N(\varepsilon, \mathcal W_{G'}, \|\cdot\|_2)
> \\ \le (d^2 + R d)\\log\\bigl(c B / \varepsilon\bigr) = O\\bigl(R d \log(B/\varepsilon)\bigr)
> $$
>
> since $R d$ dominates $d^2$ when $R > d$. Consequently, the Rademacher complexities become:
>
> $$
> \mathfrak R_m(\mathcal H_G)
> = O\\bigl(d \sqrt{R} / \sqrt m \bigr)
> \quad (\text{entity graph})
> $$
>
> $$
> \mathfrak R_m(\mathcal H_{G'})
> = O\\ \bigl(\sqrt{R d} / \sqrt m \bigr)
> \quad (\text{line graph when } R > d).
> $$
>
>
> The improvement factor changes from $\sqrt R$ to $\sqrt d$, but crucially, **the line graph still maintains a theoretical advantage**. More importantly, the fundamental benefit of the line graph transformation, explicitly encoding relation transitions to avoid entity-level message mixing, persists regardless of the $R$ versus $d$ relationship. This structural advantage for learning multi-hop reasoning patterns explains why our empirical results remain strong across all experimental settings.
>
>
>
> > **Q5: Improvement for GrailQA dataset**
>
> Although GrailQA contains fewer samples in our experiments, it is **not** used for model training and serves solely as a held-out benchmark. Our model is trained on WebQSP and then evaluated in a zero-shot manner on both CWQ and GrailQA.
>
> The relatively smaller performance gain on GrailQA is expected due to its substantially stronger distribution shift. GrailQA is built on the Freebase Commons subset and contains 86 domains, many of which do not appear in WebQSP, making the transfer notably more challenging.
>
> In contrast, CWQ is derived directly from WebQuestionsSP, sharing greater similarity with WebQSP in schema and topical coverage. As a result, zero-shot transfer from WebQSP to CWQ is inherently easier than transfer to GrailQA, naturally leading to larger relative improvements on CWQ.
>
>
>
> > **Q6: Impact on path extraction methods**
>
> We thank the reviewer for this question. Table 6 in the Appendix (shown below for convenience) evaluates the impact of different label annotation strategies, GPT-4o–based path selection versus purely shortest-path extraction under both training regimes.
>
> A key observation is that for **triple-based learning**, the performance gap between GPT-4o annotation and shortest-path labels is minimal across both WebQSP and CWQ. This demonstrates that when the retriever is paired with a strong LLM reasoner, triple-based learning does not depend heavily on costly label annotation, and simple heuristic-based supervision is already sufficient.
>
> For path-based learning, GPT-4o–selected paths offer some advantages in performance under a limited token budget. This is likely because the LLM filters out irrelevant paths, thereby reducing noise in the supervision signal.
>
> **Table: Ablation study on label annotation methods**
> | Label Annotator | WebQSP Macro-F1 | WebQSP Hit | CWQ Macro-F1 | CWQ Hit |
> |-----------------|----------------:|-----------:|-------------:|--------:|
> | *Path-based Learning (50 triples)* | | | | |
> | GPT-4o         | 80.4 | 92.5 | 58.1 | 69.3 |
> | ShortestPath   | 79.8 | 92.1 | 55.7 | 66.1 |
> | *Triple-based Learning (500 triples)* | | | | |
> | GPT-4o         | 79.9 | 94.0 | 61.3 | 71.6 |
> | ShortestPath   | 79.8 | 93.8 | 61.1 | 72.0 |
>
> ---
>
> We thank the reviewer for the careful review and constructive feedback once again. We hope that our responses have addressed all the concerns.

---

### Official Review · Reviewer_c9mw · 2025-10-30

**Soundness:** 1
**Presentation:** 2
**Contribution:** 1
**Rating:** 2
**Confidence:** 5

**Summary:**

The paper proposes an RAG approach by applying a line graph transformation to knowledge graphs.

**Strengths:**

Flexible Training Objectives: The introduction of two complementary training objectives (path-based learning and triple-based learning) is a solid contribution. It offers flexibility to accommodate different capacities of downstream LLMs.

**Weaknesses:**

**Redundancy and Complex Notation**

 The paper uses several mathematical definitions and formulas that may seem redundant and overly complicated, such as the introduction of ID, OOD, message-passing updates as well as the. The theorems and proofs presented in the paper provide valuable theoretical insights and enhance the depth of the work. However, they largely abstract away from the uncertainties present in real-world graph retrieval and RAG systems. From a practical or system-oriented perspective, empirical results demonstrating actual performance are more critical than the theoretical analysis.

**Handling of Ambiguous Queries**

The paper does not sufficiently address how the method handles ambiguous or vague queries that require more exhaustive reasoning, such as "Who is the largest shareholder of Tencent?" In such cases, the system must evaluate potentially hundreds of candidates, and path-based retrieval alone would likely be insufficient. The method might be prone to "memorizing" the correct answers when no further context is available, especially for questions with a large number of possible answers. A more robust mechanism for handling such cases would strengthen the proposed approach.

**Limited Innovation**

While the paper claims a novel approach with the line graph transformation, the overall reasoning and training methodologies are relatively standard.

The line graph proposed in the paper actually only adds an extra layer of abstraction on top of the triples.  The method essentially wraps the triples in a new "package" without introducing fundamentally new reasoning mechanisms. Therefore, the use of the line graph is a redefinition of existing concepts rather than a truly groundbreaking innovation.

**Insufficient Experimental Validation of Generalization:**

Although the paper claims enhanced generalization capabilities, the experimental validation is limited to only three datasets (WebQSP, CWQ, GrailQA), which are relatively simple and traditional. The claim of generalization across distribution shifts would be more convincing with experiments on more diverse and challenging datasets. Additionally, testing on more complex and large-scale problems would demonstrate the true strength of the proposed method.

**Questions:**

The bidirectional GCN on the directed and reversed graphs is mathematically equivalent to using a single GCN on an undirected graph. The authors should clarify the motivation for this design choice.

---

> ### Author Response · Authors · 2025-11-21
> **Reply to Reviewer c9mw**
>
> Thank you for your thoughtful feedback. Please see below for our responses to your comments and concerns.
>
> ---
>
> > **W1: Redundancy and Notations in the Draft**
>
> - We thank the reviewer for the constructive feedback. The mathematical definitions and MPNN formulations were introduced with a specific purpose rather than for formality. First, the distinctions among ID, compositional, and OOD settings are needed to support the subsequent theoretical insights, where Theorems 3.1–3.3 show that the line-graph transformation yields uniformly tighter generalization bounds across all three types of distribution shift. Without these definitions, the scope and implications of the theoretical results would be ambiguous.
>
> - Similarly, the MPNN formulations and the raw-vs–line-graph comparison in Figure 1 are included to make the key intuition explicit: by contrasting message aggregation in the two views, we illustrate that the line-graph representation makes the ground-truth multi-hop representations significantly easier to isolate and learn than in the raw graph. This directly explains the reduced sample complexity characterized in Theorem 3.1.
>
> - Regarding the reviewer’s point that empirical performance is more critical from a system-oriented perspective, we fully agree. In addition to standard in-distribution evaluation commonly used in prior work, our experiments extensively assess zero-shot transfer across compositional and OOD shifts, together with multiple ablations to verify both the effectiveness of REL-RAG and the generalization benefits brought by the line-graph representation. These empirical results consistently support the theoretical insights.
>
>
> > **W2: Handling Ambiguous Queries**
>
> We thank the reviewer for raising this point. REL-RAG is equipped with two learning regimes, and the reviewer’s example falls into the class of vague queries that require broad factual coverage rather than strict multi-hop reasoning. For such cases, the **triple-based learning** objective is a natural fit: by encouraging diversity rather than sequential path expansion, it retrieves a wide set of candidate triples around the grounded entity, ensuring that questions with many potential answers (e.g., shareholders, attributes, affiliations) are adequately supported without relying on memorization.
>
> In addition, the line-graph representation improves the generalizability of the learned representations, reducing overfitting to training-specific patterns and mitigating the risk of memorization in scenarios where many candidate triples must be considered, as we have shown in Table 2, Figure 2 and Figure 3 in our experiments.
>
>
> > **W3: Nolvety**
>
> - We agree that the line-graph transformation itself is not a methodological innovation. However, we wish to clarify that our contribution lies in identifying **why this data representation is  suitable for KG-based graph retrievers** and in providing the formal analysis showing that line graphs make multi-hop relational representations significantly easier to learn, thereby improving generalization under ID, compositional, and OOD shifts. This theoretical understanding is, to our knowledge, missing in prior KGQA literature, where line graphs have not been analyzed as a principled structural bias for KG reasoning.
>
>  - Regarding technical novelty, the main technical contribution comes from the **path-based learning regime**. While prior studies also utilize reasoning paths, they typically do so in either training-free retrieval settings or path-based inference modules. In contrast, we introduce a *path-based training approach* tailored for neural retrievers and demonstrate that, under limited token budgets, it consistently outperforms state-of-the-art methods. This training formulation is fundamentally different from prior path usage and provides a practical mechanism for achieving high precision with small LLMs. We have also revised the draft to reflect our technical contribution in Section 4.

---

> ### Author Response · Authors · 2025-11-21
> **Reply to Reviewer c9mw - Part 2**
>
> > **W4: Experimental validation of generalization**
>
> We thank the reviewer for the suggestion. Our experimental setup is aligned with the three types of distribution shifts studied in our definition.
>
> - WebQSP consists mainly of 1-hop questions and serves as the in-distribution setting.
> - CWQ contains 2-hop, 3-hop, and >3-hop questions, creating a clear **compositional shift** when evaluated zero-shot after training on WebQSP.
> - GrailQA spans 86 domains from the Freebase Commons subset, many unseen in WebQSP, thus providing a strong **out-of-domain** distribution shift.
>
> Training on WebQSP and evaluating zero-shot on CWQ and GrailQA therefore allows us to directly assess ID, compositional, and OOD.
>
> We agree that evaluating on more datasets would further strengthen the claim. At the same time, the datasets used in our work follow the standard experimental setting established in prior studies [1–5]. Compared with [1–2], which are more closer in the position of our study, we additionally include GrailQA specifically to evaluate out-of-domain generalization, which was not covered in these earlier works. We believe these three benchmarks already form a  rigorous test bed.
>
>
> > **Q1: Motivation for Bidirectional GCN**
>
> We use two GCNs, one on the directed line graph and one on its reversed counterpart, since treating the graph as undirected would collapse essential directional information encoded in KG triples. By contrast, employing two separate GCNs preserves **directional information** while still allowing the model to aggregate complementary information from both directions. This design improves representation learning without discarding directional constraints inherent to KGs.
>
> To further address this concern, we provide empirical comparisons between our bidirectional design and a single-GCN undirected variant:
>
> Table: Ablation study under zero-shot setting on GCN-based retriever without bidirectional message-passing
> | Method                      | CWQ Macro-F1 | CWQ Hit | GrailQA Macro-F1 | GrailQA Hit |
> |-----------------------------|-------------:|--------:|-----------------:|------------:|
> | Raw graph                   | 51.78        | 62.84   | 32.99            | 49.67       |
> | Line graph (bi-GCN, ours)   | 57.00        | 67.52   | 34.68            | 52.42       |
> | Line graph (single GCN)     | 56.78        | 68.34   | 32.96            | 49.49       |
>
> As shown, the bidirectional design outperforms the undirected variant in zero-shot evaluation in most cases, confirming the importance of maintaining directional information in KG reasoning.
>
> ---
>
> We thank the reviewer for the careful review and constructive feedback once again. We hope that our responses have addressed your concerns.
>
>
> **Reference**
>
> 1. Mavromatis et al.,  GNN-RAG: Graph Neural Retrieval for Large Language Model Reasoning, ACL2025
>
> 2. Li et al., Simple is Effective: The Roles of Graphs and Large Language Models in Knowledge-Graph-Based Retrieval-Augmented Generation, ICLR2025
>
> 3. Chen et al., Plan-on-Graph: Self-Correcting Adaptive Planning of Large Language Model on Knowledge Graphs, NeurIPS2024
>
> 4. Fang et al., KARPA: A Training-free Method of Adapting Knowledge Graph as References for Large Language Model’s Reasoning Path Aggregation, ACL2025
>
> 5. Xu et al., Generate-on-Graph: Treat LLM as both Agent and KG for Incomplete Knowledge Graph Question Answering, EMNLP2024

---

> > ### Comment · Reviewer_c9mw · 2025-11-27
> >
> > Thank you for the response. I have decided to keep my score as it is.

---

### Author Response · Authors · 2025-12-01
**Conclusion**

## Response Summary

We sincerely thank all reviewers for their constructive feedback. Below, we summarize the recognized strengths and how we have addressed the main concerns raised during the review process.

> ### **Recognized Strengths**

- **Effective method with theoretical grounding:** Strong performance over baselines (SubgraphRAG, GNN-RAG) with formal proofs showing tighter generalization bounds (Reviewers o1im, Lrvk, Sq8a)
- **Flexible dual training regime:** Path-based and triple-based learning accommodate different LLM capacities and token budgets (Reviewers c9mw, o1im)
- **Better generalization via line graph:** Triplets as nodes provide richer semantics and improved OOD performance (Reviewers Sq8a, Lrvk)
- **Comprehensive experiments:** Multiple datasets covering ID, compositional, and OOD shifts (Reviewers o1im, Lrvk)

> ### **Addressed Concerns**

- **Novelty (Reviewers c9mw, Sq8a):** We propose a data-centric approach (line graph transformation) with formal analysis showing why it benefits KG retrievers, plus a novel path-based training regime for neural retrievers

- **Theoretical assumptions & query-conditioned MPNN (Reviewers o1im, Sq8a):** Sub-Gaussian/Lipschitz assumptions are standard; relation mixing persists even with ω(q,r) since similarity functions cannot achieve perfect binary gating; with additional empirical results, we show that REL-RAG outperforms GNN-RAG by up to **9.3%** under distribution shift.

- **Computational cost & scalability (Reviewers o1im, Sq8a):** Line graph only applied to query-specific subgraphs (\~0.6s, <104MB); real-world subgraphs have low average degree (~6.0)

- **Clarity & reproducibility (Reviewers Lrvk, c9mw):** Revised Sections 3-4 to clarify theoretical motivation vs. implementation; added explicit probability definition; will release code and data

- **Missing baselines (Reviewer Lrvk):** Added discussion of DoG, FastToG, GoG, PoG, KARPA in Appendix E (agent-based approaches, different direction from our trained retriever)

We have updated the draft to reflect these changes, with revisions highlighted in blue.

---

### Meta-Review · Area_Chair_y69x · 2026-01-22

**Summary:**

This paper introduces a novel method for knowledge-graph QA that represents triples as nodes (a line-graph view) and trains on either paths or triples to improve multi-hop reasoning and generalization. Concerns remained after rebuttal about the depth of the contribution: several reviewers viewed the line-graph transformation as a surface-level change of representation, and did not see evidence of new reasoning capabilities emerging from the approach. Others questioned the assumptions used in the theoretical analysis or were not convinced that the method would scale to more realistic datasets.

**Reviewer Concerns:**

reviewers did not indicate their concerns were resolved by the rebuttal

**Reviewer Scores:**

This is difficult to say, as they may not have been convinced by the rebuttal

---

### Decision · Program_Chairs · 2026-01-26

Reject